# DRO-AUGMENT FRAMEWORK: ROBUSTNESS BY SYNERGIZING WASSERSTEIN DISTRIBUTIONALLY ROBUST OPTIMIZATION AND DATA AUGMENTATION

## ABSTRACT

In many real-world applications, ensuring the robustness and stability of deep neural networks (DNNs) is crucial, particularly for image classification tasks that encounter various input perturbations. While data augmentation techniques have been widely adopted to enhance the resilience of a trained model against such perturbations, there remains significant room for improvement in robustness against corrupted data and adversarial attacks simultaneously. To address this challenge, we introduce DRO-Augment, a novel framework that integrates Wasserstein Distributionally Robust Optimization (W-DRO) with various data augmentation strategies to improve the robustness of the models significantly across a broad spectrum of corruptions. Our method outperforms existing augmentation methods under severe data perturbations and adversarial attack scenarios while maintaining the accuracy on the clean datasets on a range of benchmark datasets, including but not limited to CIFAR-10-C, CIFAR-100-C, Tiny-ImageNet-C, and Fashion-MNIST. On the theoretical side, we establish novel generalization error bounds for neural networks trained using a computationally efficient, variation-regularized loss function with augmented data, closely related to the W-DRO problem. Furthermore, we introduce a refined CIFAR-C benchmark that corrects inconsistencies in corruption intensities, providing a more reliable evaluation for future robustness research.

## 1 INTRODUCTION

Deep Neural Networks (DNNs) have become essential tools in fields such as computer vision, natural language processing, speech recognition, and autonomous systems. Their ability to model complex, non-linear relationships in large-scale datasets has led to significant progress in both research and real-world applications. By achieving state-of-the-art performance in tasks like image classification (Krizhevsky et al., 2012; He et al., 2016a), speech recognition (Hinton et al., 2012; Chan et al., 2016), natural language processing (Vaswani et al., 2017; Devlin et al., 2019), DNNs have established themselves as a core component of modern artificial intelligence.

Despite their (super)-human performance on clean datasets, DNNs are often found to be highly sensitive to noisy or adversarial inputs (Szegedy et al., 2013; Gilmer et al., 2018) Various studies have demonstrated that models trained on unperturbed data can experience significant performance drops when tested on corrupted datasets (Rusak et al., 2020), such as CIFAR-10-C, CIFAR-100-C, Tiny-ImageNet-C(Hendrycks & Dietterich, 2019) (which are corrupted versions of the CIFAR-10, CIFAR-100 and Tiny-ImageNet datasets, respectively), where typically the corruption is introduced by blurring, adding noise to the images, and in various other natural ways. Additionally, adversarial attacks, like Projected Gradient Descent (PGD), (Mądry et al., 2017), AutoAttack (Croce & Hein, 2020b), C&W (Carlini & Wagner, 2017), FAB-T (Croce & Hein, 2020a) and Square Attack (Andriushchenko et al., 2020) generate small, carefully designed perturbations that can cause DNNs to make incorrect predictions with high confidence. These vulnerabilities are particularly concerning in critical applications such as autonomous driving, healthcare, and security, where model failures could have serious consequences on human life or our society as a whole.

To alleviate these issues, there is a growing interest in developing models that are more robust against various natural perturbations. One key approach for achieving this is to augment the training data

with a variety of perturbations during the training of the model. Methods like Mixup (Zhang, 2017), AugMix (Hendrycks et al., 2019) and NoisyMix (Erichson et al., 2022) aim to increase the diversity of training samples for exposing the model to a broader range of input variations. This exposure helps DNNs to generalize better to unseen perturbations and enhances their robustness against natural corruptions.

Another promising line of research aims to achieve robustness by optimizing a robust loss function. One of the most popular approaches in this area is Wasserstein distributionally robust optimization (W-DRO) (Delage & Ye, 2010; Mohajerin Esfahani & Kuhn, 2018), which trains a model by minimizing the worst-case expected loss over a Wasserstein ball (the radius is denoted by $\rho$) centered at the empirical distribution of the training samples:

$$\min_{\beta} \sup_{\mathbb{P}:W_p(\mathbb{P},\mathbb{Q})\leq\rho} \mathbb{E}_{z\sim\mathbb{P}}[\ell(f_\beta(z))], \tag{P}$$

where $f$ is the prediction function, $\mathbb{Q}$ is a nominal distribution, $\beta$ is a parameter vector to be learned, and $\ell \in \mathcal{L}$ represents the loss function dependent on the random data $z$. Notable studies include robust regression models in (Blanchet et al., 2019a; Chen & Paschalidis, 2018), adversarial training for neural networks in (Sinha et al., 2017a), and distributionally robust logistic regression in (Shafieezadeh Abadeh et al., 2015; Chen et al., 2021). A growing body of work has shown that W-DRO effectively penalizes a certain norm of the predictor's gradient, leading to a formal connection between the standard W-DRO formulation (Equation equation P) and the penalized version, known as variational regularization (see (Blanchet et al., 2019b; Chen & Paschalidis, 2018; Mohajerin Esfahani & Kuhn, 2018; Bartl et al., 2020; Blanchet et al., 2019b; 2022) for details). Recently, a general theory regarding the relation between this variation regularization and W-DRO has been developed in (Gao et al., 2024), which accommodates non-convex and non-smooth loss functions. W-DRO has been shown to be effective against adversarial perturbations (Chen et al., 2021; Bai et al., 2023; Ho-Nguyen & Wright, 2023).

Although data augmentation methods have demonstrated effectiveness against certain types of corruptions (e.g., blur, glass), our experimental results show that they remain vulnerable to others, particularly adversarial attacks. W-DRO has been shown to improve robustness against adversarial perturbations by explicitly accounting for distributional uncertainty during training. However, its impact, especially on the accuracy under natural corruptions, remains comparatively underexplored in the existing literature. Therefore, it is desirable to develop methods that are robust to both natural and adversarial corruptions.

Motivated by the complementary strengths of these two approaches, we propose DRO-Augment, a novel framework that integrates DRO with advanced data augmentation techniques to bring the best of both worlds. DRO-Augment is a training procedure that combines data augmentation methods with distributionally robust optimization by regularizing the gradient of the loss, aiming to enhance the model's resilience to both extreme data perturbations and adversarial scenarios. Unlike standard augmentation methods, which primarily focus on improving robustness against specific types of perturbations, DRO-Augment further leverages DRO's capacity to optimize worst-case distributions, ensuring improved robustness across a broader spectrum of distortions, without sacrificing accuracy.

Our contributions can be summarized as follows:

1. **Novel Framework:** We propose DRO-Augment, a theoretically motivated training framework that integrates the strengths of DRO and advanced data augmentation techniques to enhance robustness against both common corruptions and adversarial attacks.
2. **Theoretical analysis:** We present a theoretical analysis of our proposed methodology, demonstrating the regularization effects of DRO-Augment. Moreover, we establish a generalization error bound for the combination of W-DRO and augmented data using deep neural networks, thereby providing theoretical guarantees on the generalization performance of DRO-Augment under worst-case distributional shifts.
3. **Empirical Evaluation:** Through extensive experiments on benchmark datasets (including CIFAR-10-C, CIFAR-100-C, Tiny-ImageNet-C, and Fashion-MNIST), we demonstrate that DRO-Augment significantly outperforms existing methods in terms of accuracy under severe perturbations and adversarial attacks, while maintaining performance on the clean dataset.
4. **Refined Corruption Benchmark:** We identify inconsistencies and limited discriminative problem in the original CIFAR-C severity settings. To address this, we redefine the severity

scale by calibrating it based on ResNet performance in E. This refinement enables more meaningful and consistent evaluation of robustness improvements.

## 2 METHOD

Our DRO-Augment framework relies on two pillars: W-DRO and data augmentation. W-DRO aims to improve robustness against adversarial attacks by guarding against the worst-case distribution shift. Data augmentation methods, on the other hand, enhance model robustness against common corruptions by applying transformations to input images. In our framework, we first apply a chosen data augmentation method to the training data and then minimize a Wasserstein distributionally robust loss function on the augmented samples to obtain the final predictor. Our training procedure is summarized in Algorithm 1. In the following, we describe the DRO-Augment procedure in detail.

At each epoch, we begin by applying data augmentation to the training minibatch to enhance data diversity. A range of augmentation strategies has recently gained popularity for improving robustness in prediction tasks. Our framework is flexible and can accommodate any standard augmentation technique. In this work, we focus on three representative methods: Mixup (Zhang, 2017), a widely used technique that generates new training samples by linearly interpolating between pairs of examples and their labels; AugMix (Hendrycks et al., 2019), another widely used method that combines diverse augmentation operations (such as translation and contrast adjustment) with consistency regularization via a Jensen–Shannon divergence (JSD) loss, which is defined as the average of KL divergences between each distribution and their mean; and NoisyMix (Erichson et al., 2022), a state-of-the-art approach that extends feature-level mixup with noise injection and incorporates stability training (Zheng et al., 2016) to further enhance robustness.

Next, rather than minimizing the standard empirical loss over the augmented data, we optimize the training objective using a W-DRO framework. This allows the model to be optimized against worst-case perturbations within a Wasserstein ball around the empirical distribution, providing stronger guarantees under distribution shifts. Given any predictor $f$, the Wasserstein distributionally robust loss $D_{P_n,\rho}(f)$ is defined as:

$$D_{P_n,\rho}(f) = \sup_{\mathbb{P}:W_p(\mathbb{P},\mathbb{P}_n)\leq\rho} \mathbb{E}_{(x,y)\sim P}[\ell(f(x,y))],$$

where $W_p$ is the $L_p$ Wasserstein distance and $\mathbb{P}_n$ is the empirical distribution of $\{(X_i, Y_i)\}$. While this W-DRO objective provides robustness guarantees under distribution shifts, it is generally hard to optimize directly due to the inner supremum of the Wasserstein ball constraint. To address this, we adopt a variation-regularization-based approximation of the W-DRO objective, following the approach proposed by (Gao et al., 2024), which approximates the supremum with a gradient-norm-based penalty. This leads to the following (approximate) loss function $R_n(f)$ of $D_{P_n,\rho}(f)$:

$$R_n(f) = \mathbb{E}_{P_n}[\ell(f(x,y))] + \rho \left( \tfrac{1}{n} \sum_{i=1}^n \|\nabla\ell(f(x_i,y_i))\|_*^q \right)^{\frac{1}{q}}, \tag{2.1}$$

where $\rho$ now serves as a penalty weight that controls the strength of the variation regularization. The following proposition, (Theorem 1 of (Gao et al., 2024)), provides a theoretical justification for this approximation:

**Proposition 2.1** (Theorem 1 in (Gao et al., 2024)). *Let $z = (x,y) \in \mathcal{Z} = \mathcal{X} \times \mathcal{Y}$. If the gradient and Hessian norms of $f$ are well bounded, for many important applications, it can be shown that when $\rho = O(1/\sqrt{n})$, the following asymptotic equation holds:*

$$\min_f \ D_{P_n,\rho}(f) = \min_f \left\{ \mathbb{E}_{P_n}\left[\ell(f(x,y))\right] + \rho \left( \tfrac{1}{n} \sum_{i=1}^n \|\nabla\ell(f(x_i,y_i))\|_*^q \right)^{1/q} + O(n^{-1}) \right\}.$$

This approximation makes it feasible to compute the W-DRO objective in practice. In our implementation, model parameters are updated using stochastic gradient descent (SGD) based on the regularization function $R_n(f)$. Over multiple epochs, the model iteratively refines its parameters to minimize both empirical risk and the regularization penalty, aiming to balance data fitting and model complexity.

---

**Algorithm 1** Training with DRO-Augment

---

**Require:** Training data $\mathcal{D}$, neural network model $f_\theta$, objective function $\mathcal{L}$, augmentation function $\mathcal{A}$, training epochs $N$, learning rate $\eta$, regularization weight $\rho$.

    Initialize model parameters $\theta$;

    **for** epoch $= 1$ to $N$ **do**

        **for** each minibatch $\mathcal{B} \subset \mathcal{D}$ **do**

            Apply data augmentation: $\tilde{\mathcal{B}} = \mathcal{A}(\mathcal{B})$;

            **for** each input $(x_i, y_i) \in \tilde{\mathcal{B}}$ **do**

                Compute model output: $\hat{y}_i = f_\theta(x_i)$;

                Compute total loss:

$$\mathcal{L}_{total} = \sum_i \mathcal{L}(f_\theta(x_i), y_i) + \rho \mathbb{E}_{P_n}\left[\|\nabla_x \mathcal{L}(f_\theta(x_i), y_i)\|_q\right];$$

            **end for**

            Update $\theta$ using stochastic gradient descent:

$$\theta \leftarrow \theta - \eta \nabla_\theta \mathcal{L}_{total};$$

        **end for**

    **end for**

---

## 3 EXPERIMENTS

### 3.1 DATASET

To compare the performance of our method with other state-of-the-art methods, we conduct experiments on a suite of benchmark datasets, which are curated to evaluate the robustness of deep neural networks against common corruptions as well as adversarial attacks. The CIFAR-10-C, CIFAR-100-C and Tiny-ImageNet-C datasets (Hendrycks & Dietterich, 2019) are widely used to evaluate model robustness under synthetic/natural corruptions. These datasets are created by introducing 15 distinct types of corruption (including, but not limited to, noise, blur, weather effects, and digital distortions) to the CIFAR-10, CIFAR-100 and Tiny-ImageNet datasets, respectively. Each of these fifteen corruption types is applied at five predefined severity levels, which control the intensity of the corruption based on human perceptual assessment. These corrupted datasets enable a systematic analysis of model performance at various levels and types of corruption. The average classification accuracy across all corruption types and severity levels is commonly used as a metric to assess robustness (e.g., see Hendrycks & Dietterich (2019); Chen & Paschalidis (2018); Erichson et al. (2022); Zhang (2017)).

To evaluate the efficacy of our method against adversarial attacks, we use the Tiny-ImageNet and Fashion-MNIST (Xiao et al., 2017) datasets, with adversarially perturbed examples generated using the Projected Gradient Descent (PGD) attack, AutoAttack, C&W attack, FAB-T and Square attack. The PGD attack (Mądry et al., 2017) is a standard method for creating adversarial examples by iteratively applying small perturbations to the input within a bounded norm. AutoAttack (Croce & Hein, 2020b) is a robust evaluation suite that automatically combines multiple complementary attacks to yield a reliable estimate of worst-case robustness. C&W attack (Carlini & Wagner, 2017) formulates adversarial example generation as an optimization problem, aiming to find minimal perturbations that confidently change the model's prediction. FAB-T attack (Croce & Hein, 2020a) focuses on targeted attacks by approximating the decision boundary and iteratively projecting onto it to minimize $\ell_p$-norm perturbations. Square attack (Andriushchenko et al., 2020), in contrast to gradient-based methods, adopts a randomized black-box strategy using square-shaped noise patterns to explore the input space efficiently.

The Fashion-MNIST dataset contains 70,000 grayscale images of fashion items from 10 distinct categories, such as T-shirts, trousers, sneakers, etc. While MNIST(LeCun et al., 1998) serves as a classical baseline for simple visual pattern recognition tasks, Fashion-MNIST offers a more challenging alternative with a greater diversity of visual features. Tiny-ImageNet is a subset of the ImageNet dataset, containing 100,000 color images of size 64×64 across 200 classes. Each class has 500 training and 50 validation images. Compared to datasets like Fashon-MNIST or CIFAR-10, Tiny ImageNet poses a greater challenge and is commonly used to evaluate model performance

and robustness in more complex visual settings. The adversarial attacks target vulnerabilities in the model's decision boundary and provide a rigorous evaluation of adversarial robustness. Unlike natural corruptions, these adversarial examples focus on the ability of models to resist malicious, worst-case perturbations crafted to deceive classifiers. The details of adversarial experiments are provided in the Baseline and Training Details section.

To comprehensively evaluate the robustness of our proposed framework, DRO-Augment, we use the CIFAR-10-C, CIFAR-100-C and Tiny-ImageNet-C datasets to assess robustness against natural corruption, and the Fashion-MNIST and Tiny-ImageNet datasets to assess robustness against adversarial attacks.

## 3.2 BASELINE AND TRAINING DETAILS

In our experiments on the corrpution datasets, we use three different data augmentation methods: Mixup, AugMix, and NoisyMix. For each of these augmentation methods, we also apply the corresponding DRO-Augment framework and compare their performance. We used PreActResNet-18 He et al. (2016b) – a variant of ResNet that applies batch normalization and ReLU activation before each convolution – for NoisyMix, Mixup, and AugMix. We train all models for 200 epochs. We also keep consistent hyperparameter configurations (e.g., learning rate, batch size, and number of epochs) across all methods for fair comparisons in the CIFAR-10-C, CIFAR-100-C and Tiny-ImageNet-C experiments.

For the adversarial robustness experiments, we also employed PreActResNet-18 as the primary model for all methods, with training performed over 50 epochs. Let $\epsilon$ define the maximum perturbation under the $L_\infty$ -norm constraint. We generate adversarially corrupted images by using Projected Gradient Descent (PGD) (Mądry et al., 2017) with 20 iterations, enforcing $\epsilon \in \{4/255, 8/255, 16/255\}$ and using a step size of $\epsilon/8$. For the C&W attack (Carlini & Wagner, 2017), we run 10 optimization steps with the parameter $c \in \{1, 5, 10\}$. For AutoAttack (Croce & Hein, 2020b), we use the full ensemble with $\epsilon \in \{2/255, 4/255, 8/255\}$. Both the FAB-T attack (Croce & Hein, 2020a) and the Square attack (Andriushchenko et al., 2020) are performed with perturbation size fixed at $\epsilon = 8/255$. We refer to the adversarially perturbed versions Fashion-MNIST and Tiny-ImageNet as Fashion-MNIST-$\epsilon$ and Tiny-ImageNet-$\epsilon$ respectively. As in the Fashion-MNIST and Tiny-ImageNet experiments, the training parameters were kept consistent across all experiments to ensure a fair and reliable comparison.

For the experimental setup, we train our models using stochastic gradient descent (SGD) with Nesterov momentum. The momentum coefficient is set to 0.9, and the weight decay is fixed at 0.0005. The initial learning rate is set to $1 \times 10^{-1}$ and follows a cosine annealing schedule, gradually decaying to a minimum value of $1 \times 10^{-5}$ over the course of training. We use a mini-batch size of 128 for training and a batch size of 1000 during evaluation. For data augmentation methods such as AugMix, Mixup, and NoisyMix, and adversarial training baselines such as TRADES(Zhang et al., 2019), MART(Wang et al., 2019), and DRO-AT(Sinha et al., 2017b), we adopt the recommended hyperparameter values as specified in their original papers. For these adversarial training baselines as well as our proposed method DRO-Augment, the choice of perturbation budget $\epsilon$ and Wasserstein ball radius $\rho$ is typically guided by the strength of the adversarial perturbation (i.e., the chosen $\epsilon$), followed by light hyperparameter tuning to balance robustness and clean accuracy.

All experiments are conducted on local GPU workstations equipped with two NVIDIA RTX A6000 GPUs (each with 48 GB of VRAM). For CIFAR-10-C, CIFAR-100-C and Tiny-ImageNet-C , the average training time per epoch under the DRO-Augment setting is approximately 0.019 hours. For Fashion-MNIST, each epoch completes within a few seconds due to the smaller input size and model complexity. The full implementation and all experiment scripts are publicly available at https://anonymous.4open.science/r/DRO-Augment-6F2F.

## 3.3 ADVERSARIAL ATTACK RESULTS

In the experiments on Fashion-MNIST and Tiny-ImageNet, we trained our model using the Fashion-MNIST and Tiny-ImageNet training sets, respectively. We introduced multiple attacks perturbations with different perturbation strengths to evaluate the effectiveness of the DRO-Augmented version in defending against varying levels of adversarial attacks. Table 11 and 2 present the comparisons of

Table 1: Adversarial robustness accuracy comparison under multiple attacks with different $\epsilon$ values in Fashion-MNIST-$\epsilon$ datasets.

| Method | AA(%) | | | PGD(%) | | | C&W(%) | | | FAB-T(%) | Square(%) |
|---|---|---|---|---|---|---|---|---|---|---|---|
| | 2/255 | 4/255 | 8/255 | 4/255 | 8/255 | 16/255 | $c=1$ | $c=5$ | $c=10$ | 8/255 | 8/255 |
| Baseline | 11.10 | 10.20 | 7.70 | 11.02 | 9.66 | 7.30 | 10.27 | 10.24 | 10.24 | 92.60 | 9.20 |
| Mixup | 32.00 | 30.20 | 25.10 | 31.01 | 27.52 | 23.46 | 30.61 | 28.25 | 28.06 | 93.20 | 28.60 |
| Mixup + DRO | 40.80 | 38.30 | 35.30 | 39.63 | 35.02 | 29.82 | 36.02 | 34.65 | 34.45 | 93.00 | 34.80 |
| AugMix | 39.80 | 37.80 | 33.90 | 38.24 | 31.68 | 24.16 | 34.47 | 32.24 | 32.19 | 93.00 | 32.00 |
| AugMix + DRO | 45.30 | 40.50 | 37.00 | 45.57 | 38.82 | 28.16 | 39.26 | 38.32 | 38.23 | 93.20 | 38.80 |
| NoisyMix | 40.70 | 38.20 | 35.00 | 39.87 | 36.89 | 31.81 | 37.68 | 35.84 | 35.74 | **93.40** | 35.40 |
| NoisyMix + DRO | **46.70** | 43.50 | **41.50** | **47.97** | **43.97** | 37.34 | **42.86** | **41.71** | **41.65** | 93.20 | **42.00** |
| TRADES | 46.15 | **44.10** | 39.90 | 45.32 | 41.96 | 33.49 | 42.08 | 41.18 | 41.09 | 91.60 | 41.40 |
| DRO-AT | 45.25 | 43.24 | 39.30 | 44.26 | 40.16 | 33.48 | 40.24 | 39.37 | 39.31 | 91.35 | 40.90 |
| MART | 28.30 | 27.00 | 25.00 | 28.69 | 26.47 | 22.60 | 27.23 | 25.82 | 25.60 | 91.70 | 25.95 |

Table 2: Adversarial robustness accuracy comparison under multiple attacks with different $\epsilon$ values on Tiny-ImageNet-$\epsilon$ dataset.

| Method | AA(%) | | | PGD(%) | | | C&W(%) | | | FAB-T(%) | Square(%) |
|---|---|---|---|---|---|---|---|---|---|---|---|
| | 2/255 | 4/255 | 8/255 | 4/255 | 8/255 | 16/255 | $c=1$ | $c=5$ | $c=10$ | 8/255 | 8/255 |
| Baseline | 4.81 | 2.45 | 0.79 | 3.69 | 1.54 | 0.52 | 1.50 | 1.10 | 1.03 | 48.15 | 2.29 |
| Mixup | 4.92 | 2.41 | 0.96 | 3.85 | 1.75 | 0.63 | 1.38 | 1.18 | 1.18 | 48.46 | 2.40 |
| Mixup + DRO | 6.18 | 3.38 | 1.27 | 5.09 | 2.36 | 0.89 | 2.08 | 1.82 | 1.80 | 50.83 | 3.21 |
| AugMix | 6.27 | 2.74 | 0.84 | 5.68 | 2.34 | 1.12 | 1.90 | 1.51 | 1.50 | 54.16 | 4.26 |
| AugMix + DRO | 7.79 | 4.01 | 1.57 | 7.39 | 3.72 | 2.68 | 3.05 | 2.71 | 2.20 | 56.56 | 5.38 |
| NoisyMix | 10.18 | 5.36 | 1.95 | 9.17 | 4.58 | 1.82 | 4.13 | 3.20 | 3.15 | 58.24 | 6.44 |
| NoisyMix + DRO | **12.55** | **7.48** | 3.65 | **11.52** | 6.80 | 3.35 | **8.19** | **6.66** | **6.54** | **59.75** | **8.18** |
| TRADES | 7.45 | 6.60 | **5.45** | 7.40 | 6.04 | 4.00 | 5.61 | 4.86 | 4.75 | 49.40 | 5.75 |
| DRO-AT | 7.80 | 6.65 | 4.95 | 7.36 | 5.97 | 3.94 | 5.60 | 4.63 | 4.47 | 50.90 | 5.60 |
| MART | 9.18 | 7.03 | 4.40 | 10.18 | **8.03** | **5.13** | 7.14 | 5.47 | 5.19 | 50.60 | 7.25 |

accuracy on the corrupted datasets between adversarial training baselines, different data augmentation methods and their corresponding DRO-Augmented versions. On Fashion-MNIST-$\epsilon$, the DRO-Augmented version demonstrates a notable increase in robustness across different values of $\epsilon$, without degrading standard classification accuracy, achieving an average improvement of 8% compared to the original methods. Furthermore, across both Fashion-MNIST-$\epsilon$ and Tiny-ImageNet-$\epsilon$, the DRO-Augmented variants frequently attain the highest or near-highest adversarial robustness in the majority of tested configurations, demonstrating their superior and stable defense capabilities against adversarial attacks.

### 3.4 COMMON CORRUPTION RESULTS

As in the previous subsection, we first train the PreActResNet-18 model on the clean CIFAR-10, CIFAR-100 and Tiny-ImageNet datasets using various data augmentation methods, along with their corresponding DRO-Augmented versions. We then measure the accuracy of these methods on corrupted datasets (CIFAR-10-C, CIFAR-100-C and Tiny-ImageNet-C).

The comparison of the results under severity level 5 for different corruption types in CIFAR-10-C, CIFAR-100-C and Tiny-ImageNet-C are presented in the tables 4, 5 and 9. Tables 6, 7 and 8 also show a comparison of the average results across all severity levels for different corruption types in CIFAR-10-C, CIFAR-100-C and Tiny-ImageNet-C. Table 13 provides a comparison of the overall average results across all severity levels and corruption types, further highlighting the performance differences between the standalone data augmentation methods and their corresponding DRO-Augmented versions. The tables 4, 5, 9, 6, 8, 7 and 13 are collected in Appendix D.

While the DRO-Augmented method consistently achieves higher accuracy on corrupted datasets compared to the corresponding data augmentation method across nearly all corruption types and levels (recall there are fifteen different corruption types, each with five severity levels), the improvement is particularly pronounced for the following seven corruption types: White, Shot, Impulse, Defocus, Glass, Motion, and Zoom. On these corruptions, accuracy typically improves by 2%–5%, peaking at 12.7%, with a median gain of 3.1% across all three augmentation baselines (Mixup, AugMix, NoisyMix). Notably, this robustness boost comes without sacrificing average performance. DRO-

Augment still improves overall accuracy around 1.2% in the CIFAR-10-C, CIFAR-100-C and Tiny-ImageNet-C datasets. Figures 1 illustrates the comparison of accuracy for the seven different corruption types mentioned above under the highest severity level (level = 5), highlighting the enhanced performance of DRO-Augmented methods over the standard augmentation methods. .

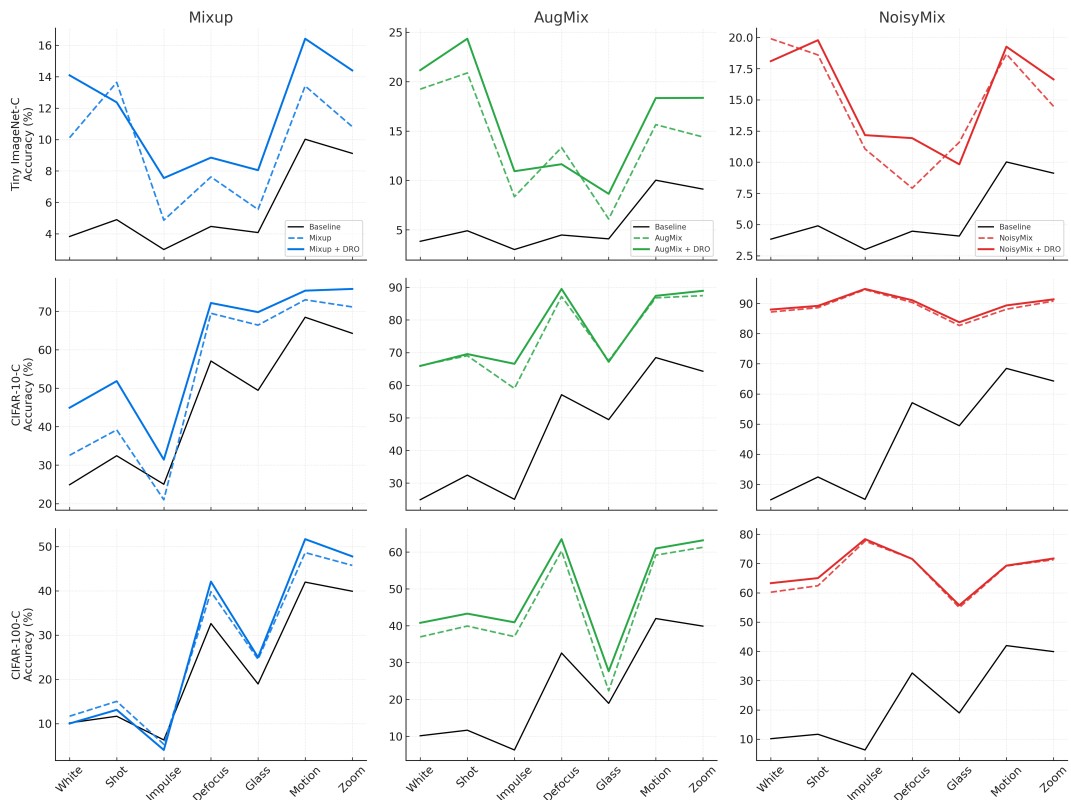

Figure 1: Accuracy comparison against the seven corruption types at severity 5 for Tiny ImageNet-C, CIFAR-10-C and CIFAR-100-C (rows). Columns show Mixup, AugMix and NoisyMix with / without DRO. Solid lines = method + DRO; Dashed lines = original method; Black line = baseline.

### 3.5 ABLATION STUDY

In this subsection, we present our findings from an ablation study aimed at understanding the impact of each component, including Mixup, standard data augmentation, JSD loss, and W-DRO regularization, on the accuracy of the trained model on corrupted datasets. Towards that goal, we select CIFAR-100-C and Fashion-MNIST-$\epsilon$ as the test datasets for common corruption and adversarial attack experiments, respectively. As before, we train PreAct-ResNet18 on the clean datasets (CIFAR-100 and Fashion-MNIST), assessing its robustness on the corresponding corrupted versions (CIFAR-100-C and Fashion-MNIST-$\epsilon$ with $\epsilon = 8/255$). Our results are presented in Table 3, which demonstrates that W-DRO significantly improves robustness against both common corruptions and adversarial attacks when combined with various augmentation strategies. The accuracy on CIFAR-100-C increases $\sim 1.2\%$ and on Fashion-MNIST-$\epsilon$ increases $\sim 6\%$. This highlights the advantage of combining DRO with data augmentation methods to achieve higher accuracy across a broad range of corruptions, including adversarial attacks.

## 4 ADVERSARIAL RISK BOUND FOR NEURAL NETWORKS

In this section, we present the asymptotic generalization error bound for a neural-network-based estimator obtained via optimizing the variation-regularized loss function $R_n(f)$ (Equation equation 2.1)

Table 3: Ablation study using a Preact-ResNet18 trained on CIFAR-100 and Fashion-MNIST-$\epsilon$. The combination of W-DRO and data mixing on top of a stability training scheme on an augmented dataset boosts both robust corruption and adversarial accuracy.

| Augmented Data | Mixing | JSD Loss | W-DRO | CIFAR-100-C (%) | Fashion-MNIST-$\epsilon$ : $\epsilon = \frac{8}{255}$ (%) |
|---|---|---|---|---|---|
| × | × | × | × | 47.88 | 9.36 |
| × | × | × | ✓ | **49.24** | **16.24** |
| ✓ | × | × | × | 55.24 | 38.12 |
| ✓ | × | × | ✓ | **56.91** | **43.79** |
| × | ✓ | × | × | 52.62 | 27.52 |
| × | ✓ | × | ✓ | **54.08** | **35.02** |
| ✓ | × | ✓ | × | 57.90 | 43.57 |
| ✓ | × | ✓ | ✓ | **59.36** | **49.72** |
| ✓ | ✓ | × | × | 58.57 | 22.91 |
| ✓ | ✓ | × | ✓ | **59.81** | **31.68** |
| ✓ | ✓ | ✓ | × | 61.28 | 31.68 |
| ✓ | ✓ | ✓ | ✓ | **62.82** | **38.82** |

**with augmented data**. As illustrated in Section 2, $R_n(f)$ serves as a proxy of the W-DRO and the approximation error depends on $\rho$. Since our method demonstrates strong performance against $L_\infty$ adversarial attacks, we primarily focus on the variation regularization-based approximation of the $L_\infty$-Wasserstein DRO optimization problem, i.e. in our case,

$$R_n(f) = \frac{1}{n} \left\{ \sum_i \ell(f(x_i), y_i) + \rho \|\nabla \ell(f(x_i), y_i)\|_2 \right\}$$

Generalization error bounds for neural networks are crucial for understanding how well a trained model will perform on unseen data, providing theoretical guarantees that guide reliable deployment in practice. In recent years, there has been a surge of research focused on generalization error bounds for structured neural networks in nonparametric regression and classification settings (e.g., see Kohler & Langer (2021); Schmidt-Hieber (2020); Bhattacharya et al. (2024) and references therein). To set up notations, given any $L \in \mathbb{N}$ and a vector $\mathbf{p} = (p_0, p_1, \cdots, p_{L+1})$, a neural network with depth $L$ and width vector $\mathbf{p}$ is defined as:

$$f(x) = W_L \circ \sigma \circ W_{L-1} \circ \sigma \circ \cdots \circ \sigma \circ W_0 x,$$

where $W_i \in \mathbb{R}^{p_{i+1} \times p_i}$, and each layer includes corresponding bias vectors $v_h \in \mathbb{R}^{p_{h+1}}$. The activation function $\sigma$ is applied to each component of the input vector, which is taken to be $\sigma(x) = (\max\{0, x\})^2$ in our analysis. In practice, our estimator is trained not only with the W-DRO but also with augmented samples generated by mixup. For any two sample $(X_i, Y_i), (X_j, Y_j)$ (generated from a distribution, namely $P_{\text{true}}$), and $\lambda_{ij} \sim \text{Beta}(\alpha, \beta)$, we define their mixture as $Z_{ij} = \lambda_{ij}(X_i, Y_i) + (1 - \lambda_{ij})(X_j, Y_j)$. The population distribution of $Z_{ij}$ is denoted as $P_{\text{mix}}$. To preserve independence within the mixed sample, we construct the mixup dataset using disjoint pairs. Specifically, with a slight abuse of notation, we define $Z_i = \lambda_i (X_{2i-1}, Y_{2i-1}) + (1 - \lambda_i)(X_{2i}, Y_{2i})$, for $1 \le i \le n/2$, where we assume $n$ is even; if $n$ is odd, the last sample is discarded. Let $R_n^{mix}(f)$ denote the empirical risk (counterpart of $R_n(f)$) computed using the mixup samples $\{Z_i\}_{1 \le i \le n/2}$. Our estimator $\hat{f}_{NN}^{mix}$ is defined as:

$$\hat{f}_{NN}^{mix} = \arg\min_{f \in \mathcal{NN}_{U,L,\mathbf{p}}^{a_1, a_2}} R_n^{mix}(f). \tag{4.1}$$

where $\mathcal{NN}_{U,L,\mathbf{p}}^{a_1, a_2}$ is the collection of all neural networks with depth $L$, width vector $\mathbf{p}$, total number of active weights $U = \sum_{h=0}^{L} (\|W_h\|_0 + \|v_h\|_0)$, norm of gradient is almost $a_1/2$ and (operator) norm of the Hessian is almost $a_2/2$. The robust generalization error of $f$ with respect $L_\infty$ W-DRO loss is defined as:

$$D_{P_{\text{true}}, \rho}(f) = \sup_{Q: W_\infty(P_{\text{true}}, Q) \le \rho} \mathbb{E}_{(x,y) \sim P_{\text{true}}}[\ell(f(x), y)]$$

$$= \mathbb{E}_{(x,y) \sim P_{\text{true}}} \left[ \sup_{\tilde{x}: \|x - \tilde{x}\|_2 \le \rho} \ell(f(\tilde{x}), y) \right]$$

We define $D_{P_{\mathrm{mix}},\rho}(f)$ as the robust generalization error of $f$ with respect to $P_{\mathrm{mix}}$, and define $f_*^{mix}$ to be the minimizer of $D_{P_{\mathrm{mix}},\rho}(f)$ over class of Hölder function $\mathcal{H}^\alpha(\mathbb{R}^d)$, defined as follows:

$$\mathcal{H}^\alpha(\mathbb{R}^d) = \left\{ f : \mathbb{R}^d \to \mathbb{R} \;\middle|\; \max_{\|\mathbf{s}\|_1 \leq r} \sup_{x \in \mathbb{R}^d} |\partial^{\mathbf{s}} f(x)| \leq 1, \; \max_{\|\mathbf{s}\|_1 = r} \sup_{x_1 \neq x_2} \frac{|\partial^{\mathbf{s}} f(x_1) - \partial^{\mathbf{s}} f(x_2)|}{\|x_1 - x_2\|_\infty^\beta} \leq 1 \right\}$$

The goal of the generalization bound is to provide an upper bound on the difference between $D_{P_{\mathrm{mix}},\rho}(\hat{f}_{NN}^{mix})$ and $D_{P_{\mathrm{mix}},\rho}(f_*^{mix})$, which we present in the following theorem:

**Theorem 4.1.** *Assume that the population minimizer $f_*^{mix} \in \mathcal{H}^\alpha(\mathbb{R}^d)$ with $\alpha > d/2$. For any fixed $Z > 0$, if we have $O(\log d + \lfloor \alpha \rfloor)$ layers, with the $\max_i p_i = O(p_{L+1} \vee d(Z + \lfloor \alpha \rfloor)^d)$ and $O(p_{L+1}d(d + \alpha)(Z + \lfloor \alpha \rfloor)^d)$ non-zero weights taking their values in [-1,1], then the optimizer $\hat{f}_{NN}^{mix}$, as defined in Equation equation 4.1, with $\ell$-Lipschitz loss function, satisfies that there exists $c, \bar{\rho} > 0$ such that for all $\rho < \bar{\rho}$, with probability at least $1 - n^{-c}$,*

$$\left| D_{P_{mix},\rho}(\hat{f}_{NN}^{mix}) - D_{P_{mix},\rho}(f_*^{mix}) \right| \leq C_1 \left( \sqrt{\frac{\log n \cdot (U + \log U)}{n}} + U^{-\alpha/d} + \rho\sqrt{\frac{\log n}{n}} + \rho^2 \right).$$

*If we further select $U \asymp (n/\log n)^{\frac{d}{2\alpha+d}}$, then we have*

$$\left| D_{P_{mix},\rho}(\hat{f}_{NN}^{mix}) - D_{P_{mix},\rho}(f_*^{mix}) \right| \leq C_2 \left( \left(\frac{\log n}{n}\right)^{\frac{\alpha}{2\alpha+d}} + \rho\sqrt{\frac{\log n}{n}} + \rho^2 \right).$$

*where $c_1, c_2, C_1, C_2$ are fixed constants, depending on $(\alpha, d, a_1, a_2)$.*

**Remark 4.2.** *Although recently Liu et al. (2024) established a similar bound on a neural network-based estimator under $L_\infty$ perturbation, our analysis differs from theirs in various aspects; first, we obtain $\hat{f}_{NN}^{mix}$ by minimizing the variation approximated loss with augmented data $R_n^{mix}(f)$, whereas they did not consider the augmentation nor the variational approximation. Secondly, the class of neural networks considered in our theoretical analysis differs from them: we focus on sparsely connected networks, where sparsity is imposed through conditions on $U$, which can be implemented via dropout in practice, in contrast to the fully connected networks studied in their paper. Additionally, we employ the ReQU activation function, rather than the ReLU activation function used in their paper. As a result of these differences, we are able to achieve a faster rate of convergence: our estimator attains a convergence rate of $n^{-\frac{\alpha}{2\alpha+d}}$, compared to their rate $n^{-\frac{\alpha}{3\alpha+2d}}$. A full derivation of the theoretical bound is provided in Appendix C, where we also include the approximation formulation for the DRO-Augment objective.*

**Remark 4.3.** *In our theoretical development, we assume that mixing is performed on independent pairs, ensuring that the resulting mixed samples remain independent. In practice, however, one might consider forming all possible pairs, which would introduce correlations among the mixed samples. Our framework can be extended to this setting using empirical process techniques for U-statistics. Nevertheless, such an extension offers no additional insight beyond more cumbersome mathematical bookkeeping, and we therefore omit it here.*

## 5 CONCLUSION AND DISCUSSION

In our research, by combining the regularization effect of distributionally robust optimization with data augmentation methods, we enhance the model's robustness against various corruptions and adversarial attacks in the field of computer vision classification. This approach allows the model to better handle perturbations, resulting in improved performance and reliability when faced with different types of data distribution changes. Despite its effectiveness, DRO-Augment introduces a small additional time costs due to the evaluation of the robust loss, which is not a fundamental limitation and can be mitigated through engineering or numerical improvements. To support more meaningful robustness evaluation, we also propose a refined version of the CIFAR-C benchmark that ensures corruption strength is consistent across different corruption types at each severity level. Future work will focus on exploring the potential of variation regularization in other models, such as diffusion models and large language models (LLMs). By investigating how variation regularization can be integrated into these models, we aim to further enhance their robustness and adaptability, expanding the applicability of our methods to a broader range of machine learning architectures and applications.

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

## A  NOTATIONS

Scalars and individual data points are denoted by lowercase letters (e.g., $x$, $y$). The input-output pair is denoted by $z = (x, y)$. Prediction functions are written as $f(x)$, with associated loss $\ell(f(x), y)$ and input gradient $\nabla\ell(f(x), y)$. The true data distribution is $P_{\text{true}}$ and the empirical distribution over $n$ samples is $P_n$. Calligraphic letters (e.g., $\mathcal{H}^\alpha$, $\mathcal{F}$) denote function classes or sets. We use $\|\cdot\|_{q^*}$ for the dual norm of $\ell_q$.

## B  AUXILIARY LEMMAS

**Lemma B.1** ((Gao et al., 2024))**.** *Under the assumptions that the data distribution satisfies a bounded density condition around the set of all nondifferentiable points of $f$ : $D_f$, there are $\bar\rho, J > 0$ such that for all $\rho < \bar\rho$, with probability at least $1 - e^{-t}$, for every $f$ from a hypothesis family $\mathcal{F}$:*

$$D_{P_n,\rho}(f) = \min_{f \in \mathcal{F}} \left\{ \frac{1}{n} \sum_{i=1}^{n} \ell\big(f(x_i), y_i\big) + \rho\, \mathbb{E}_{\mathbb{P}_n}\Big[ \ell'\big(f(x), y\big) \big\| \nabla f(x) \big\|_* \Big] \right\}$$

$$+ \rho\sqrt{\frac{t}{2n}} + \rho^2(J + \|H\|_{\mathbb{P}_{n,1}}) + 2\rho\, \mathbb{E}\big[\mathcal{R}_n(\mathcal{J}_\rho)\big]$$

*where $\mathcal{J}_{\rho,\mathcal{F}} := \{x \mapsto 1\{d(x, D_f) < \rho\} : f \in \mathcal{F}, D_f \neq \emptyset\}$ and $H(x)$ is an upper bound on the operator norm of the Hessian of $f(x)$.*

**Lemma B.2** ((Zhang et al., 2020)). *Suppose the function class $\mathcal{F}$ is defined over $\mathcal{X}$ and satisfies $\sup_{f \in \mathcal{F}} \|f\|_\infty \leq D$. For any samples $x_1, \ldots, x_n$ from $\mathcal{X}$, we have*

$$\mathbb{E}_\sigma \left[ \sup_{f \in \mathcal{F}} \frac{1}{n} \sum_{i=1}^{n} \sigma_i f(x_i) \right] \leq \inf_{\delta \geq 0} \left\{ 4\delta + 12 \int_\delta^D \sqrt{\frac{\log \mathcal{N}(u, \mathcal{F}, L_2(P_n))}{n}} \, du \right\}$$

*where $\sigma = (\sigma_1, \ldots, \sigma_n)$ are i.i.d. Rademacher variables and $L_2(P_n)$ denotes the data-dependent $L_2$ metric.*

**Lemma B.3** ((Belomestny et al., 2023), Theorem 1). *Fix $\alpha > 2$ and $p, d \in \mathbb{N}$. Then, for any function $f : [0, 1]^d \to \mathbb{R}^p$ with $f \in \mathcal{H}^\alpha([0, 1]^d)$, for any integer $Z \geq 2$, there exists a neural network $h_f$ with ReQU activations such that it has $\mathcal{O}(\log d + \lfloor \alpha \rfloor)$ layers, at most $\mathcal{O}(p \vee d(Z + \lfloor \alpha \rfloor)^d)$ neurons per layer, and $\mathcal{O}(p(d\alpha + d^2(Z + \lfloor \alpha \rfloor)^d)$ nonzero weights in $[-1, 1]$, satisfying*

$$\|f - h_f\|_{\mathcal{H}^\ell([0,1]^d)} \leq \frac{C^{\alpha d} \alpha^\ell}{Z^{\alpha - \ell}} \quad \text{for all } \ell \in \{0, \ldots, \lfloor \alpha \rfloor\}.$$

**Lemma B.4** ((Bartlett et al., 1998), Theorem 2.1). *For any positive integers $U$, $k \leq U$, $L \leq U$, $l$, and $p$, considering a network with real inputs, up to $U$ parameters, $k$ computational units in $L$ layers, a single output unit (identity activation), and all other units with piecewise polynomial activation of degree $l$ and $p$ breakpoints, the VC-dimension satisfies*

$$VCdim(sgn(\mathcal{F})) \leq 2UL \log(2eULpk) + 2UL^2 \log(l + 1) + 2L.$$

*Furthermore, since $L, k = \mathcal{O}(U)$, for fixed $l$ and $p$:*

$$VCdim(sgn(\mathcal{F})) = \mathcal{O}(UL \log U + UL^2).$$

**Lemma B.5** ((Anthony & Bartlett, 2009), Theorem 12.2). *Let $\mathcal{F}$ be a set of real functions from a domain $\mathcal{X}$ to $[0, M]$. Let $\varepsilon > 0$ and $\mathrm{Pdim}(\mathcal{F})$ denote the pseudo-dimension. If $n \geq \mathrm{Pdim}(\mathcal{F})$, then the uniform covering number satisfies*

$$\mathcal{N}_\infty(\varepsilon, \mathcal{F}, n) \leq \left( \frac{enM}{\varepsilon \, \mathrm{Pdim}(\mathcal{F})} \right)^{\mathrm{Pdim}(\mathcal{F})}.$$

**Lemma B.6** ((Vershynin, 2018), Corollary 4.2.13). *The covering numbers of the unit Euclidean ball $B_2^d$ satisfy for any $\varepsilon \in (0, 1]$:*

$$\left( \frac{1}{\varepsilon} \right)^d \leq \mathcal{N} \left( B_2^d, \varepsilon \right) \leq \left( \frac{3}{\varepsilon} \right)^d.$$

# C  THEORY PROOF

## C.1  RISK BOUND PROOF

The following proof adapts the approach from (Liu et al., 2024) with modifications to account for W-DRO, augmented samples and the ReQu activation function. Throughout the proof below, let n denote the size of the augmented dataset.

The decomposition of the risk function is given by:

$$D_{P_{\mathrm{mix}},\rho}(\hat{f}_{NN}^{mix}) - D_{P_{\mathrm{mix}},\rho}(f_*^{mix})$$
$$= D_{P_{\mathrm{mix}},\rho}(\hat{f}_{NN}^{mix}) - D_{P_n,\rho}^{mix}(\hat{f}_{NN}^{mix}) + D_{P_n,\rho}^{mix}(\hat{f}_{NN}^{mix}) - R_n^{mix}(\hat{f}_{NN}^{mix}) + R_n^{mix}(\hat{f}_{NN}^{mix})$$
$$- R_n^{mix}(\bar{f}) + R_n^{mix}(\bar{f}) - D_{P_n,\rho}^{mix}(\bar{f}) + D_{P_n,\rho}^{mix}(\bar{f}) - D_{P_n,\rho}^{mix}(f_*^{mix}) + D_{P_n,\rho}^{mix}(f_*^{mix}) - D_{P_{\mathrm{mix}},\rho}(f_*^{mix})$$
$$\leq D_{P_{\mathrm{mix}},\rho}(\hat{f}_{NN}^{mix}) - D_{P_n,\rho}^{mix}(\hat{f}_{NN}^{mix}) + D_{P_n,\rho}^{mix}(\hat{f}_{NN}^{mix}) - R_n^{mix}(\hat{f}_{NN}^{mix}) + R_n^{mix}(\bar{f})$$
$$- D_{P_n,\rho}^{mix}(\bar{f}) + D_{P_n,\rho}^{mix}(\bar{f}) - D_{P_n,\rho}^{mix}(f_*^{mix}) + D_{P_n,\rho}^{mix}(f_*^{mix}) - D_{P_{\mathrm{mix}},\rho}(f_*^{mix})$$

where $D_{P_n,\rho}^{mix}(f)$ is the empirical counterpart of $D_{P_n,\rho}(f)$ computed on mixup samples. Therefore,

$$
\begin{aligned}
|D_{P_{\mathrm{mix}},\rho}(\hat{f}_{NN}^{mix}) - D_{P_{\mathrm{mix}},\rho}(f_*^{mix})| \leq\ & |D_{P_{\mathrm{mix}},\rho}(\hat{f}_{NN}^{mix}) - D_{P_n,\rho}^{mix}(\hat{f}_{NN}^{mix})| \\
& + |D_{P_n,\rho}^{mix}(\hat{f}_{NN}^{mix}) - R_n^{mix}(\hat{f}_{NN}^{mix})| \\
& + |R_n^{mix}(\bar{f}) - D_{P_n,\rho}^{mix}(\bar{f})| \\
& + |D_{P_n,\rho}^{mix}(\bar{f}) - D_{P_n,\rho}^{mix}(f_*^{mix})| \\
& + |D_{P_n,\rho}^{mix}(f_*^{mix}) - D_{P_{\mathrm{mix}},\rho}(f_*^{mix})|
\end{aligned}
$$

where $\bar{f} \in \mathcal{NN}_{U,L}^{a_1;a_2}$ is the approximation of $f_*^{mix}$, then we have

$$
\begin{cases}
D_{P_{\mathrm{mix}},\rho}(\hat{f}_{NN}^{mix}) - D_{P_n,\rho}^{mix}(\hat{f}_{NN}^{mix}) = B_1, \\
D_{P_n,\rho}^{mix}(\hat{f}_{NN}^{mix}) - R_n^{mix}(\hat{f}_{NN}^{mix}) = B_2, \\
R_n^{mix}(\bar{f}) - D_{P_n,\rho}^{mix}(\bar{f}) = B_3, \\
D_{P_n,\rho}^{mix}(\bar{f}) - D_{P_n,\rho}^{mix}(f_*^{mix}) = B_4, \\
D_{P_n,\rho}^{mix}(f_*^{mix}) - D_{P_{\mathrm{mix}},\rho}(f_*^{mix}) = B_5.
\end{cases}
$$

### C.1.1 BOUND FOR $B_2$ AND $B_3$

By Lemma B.1, we directly derive that for any function $f \in \mathcal{NN}_{U,L}^{a_1,a_2}$ with the ReQU activation function, the following holds with probability at least $1 - e^{-t}$,

$$
|R_n^{mix}(f) - D_{P_n,\rho}^{mix}(f)| = \rho\sqrt{\frac{t}{2n}} + \rho^2\left(J + \frac{a_2}{2}\right)
$$

Hence we can get the asymptotic bound of $B_2$ and $B_3$.

### C.1.2 BOUND FOR $B_4$

Define the approximation error by

$$
\mathcal{E}(\mathcal{H}^\alpha, \mathcal{NN}_{U,L}^{a_1,a_2}) = \|f - \bar{f}\|_{\mathcal{H}^0([0,1]^d)}
$$

There exists $\bar{f} \in \mathcal{NN}_{U,L}^{a_1,a_2}$ approximating the target function $f_*^{mix} \in \mathcal{H}^\alpha$ such that

$$
\|f_*^{mix} - \bar{f}\|_{\mathcal{H}^0([0,1]^d)} = O\left(\mathcal{E}(\mathcal{H}^\alpha, \mathcal{NN}_{U,L}^{a_1,a_2})\right).
$$

Indexing augmented samples by $t$ via $Z_t = (\tilde{X}_t, \tilde{Y}_t) \in P_{\mathrm{mix}}$ s.t. $\tilde{X}_t = \lambda_t X_{2t} + (1-\lambda_t)X_{2t-1}$, $\tilde{Y}_t = \lambda_t Y_{2t} + (1-\lambda_t)Y_{2t-1}$, where $\lambda_t \overset{\text{i.i.d.}}{\sim} \mathrm{Beta}(\alpha, \beta)$. Since $\ell$ is $L_\ell$-Lipschitz, the difference between $D_{P_n,\rho}^{mix}(f_*^{mix})$ and $D_{P_n,\rho}^{mix}(\bar{f})$ satisfies

$$
\begin{aligned}
D_{P_n,\rho}^{mix}(\bar{f}) - D_{P_n,\rho}^{mix}(f_*^{mix}) \leq\ & \frac{1}{n}\sum_{t=1}^n \left| \sup_{\|\tilde{X}_t' - \tilde{X}_t\| \leq \rho} \ell\left(\bar{f}(\tilde{X}_t'), \tilde{Y}_t\right) - \sup_{\|\tilde{X}_t' - \tilde{X}_t\| \leq \rho} \ell\left(f_*^{mix}(\tilde{X}_t'), \tilde{Y}_t\right) \right| \\
=\ & \frac{1}{n}\sum_{t=1}^n \left| \sup_{\|\tilde{X}_t' - \tilde{X}_t\| \leq \rho} \ell\left(f_*^{mix}(\tilde{X}_t'), \tilde{Y}_t\right) - \sup_{\|\tilde{X}_t' - \tilde{X}_t\| \leq \rho} \ell\left(\bar{f}(\tilde{X}_t'), \tilde{Y}_t\right) \right| \\
\leq\ & \frac{1}{n}\sum_{t=1}^n \sup_{\|\tilde{X}_t' - \tilde{X}_t\| \leq \rho} \left| \ell\left(f_*^{mix}(\tilde{X}_t'), \tilde{Y}_t\right) - \ell\left(\bar{f}(\tilde{X}_t'), \tilde{Y}_t\right) \right| \\
\leq\ & L_\ell \|f_*^{mix} - \bar{f}\|_{\mathcal{H}^0([0,1]^d)}
\end{aligned}
$$

Based on Lemma B.3, we derive

$$
B_4 \leq \frac{L_\ell C^{\alpha d}}{Z^\alpha}
$$

### C.1.3 Bound for $B_5$

For any $f \in \mathcal{H}^\alpha$ and $z = (x, y) \in \mathcal{Z}$, define

$$\tilde{\ell}(f, z) = \sup_{\|x' - x\| \leq \rho} \ell(f(x'), y).$$

We have

$$B_5 = D_{P_n, \rho}^{mix}(f_*^{mix}) - D_{P_{\mathrm{mix}}, \rho}(f_*^{mix}) \leq \sup_{f \in \mathcal{NN}_{U,L}^{a_1, a_2}} \left\{ \mathbb{E}_{P_n, \rho}\big[\tilde{\ell}(f, Z_t)\big] - \mathbb{E}_P\big[\tilde{\ell}(f, Z_t)\big] \right\}.$$

Define the class

$$\mathcal{L}^\alpha = \left\{ \tilde{\ell}(f, \cdot) : \mathcal{Z} \to \mathbb{R} \mid f \in \mathcal{H}^\alpha \right\},$$

and let the random vector $\sigma = (\sigma_1, \ldots, \sigma_n)$ consist of i.i.d. Rademacher variables that are independent of the data. Denote the samples by $Z_{1:n} = \{Z_t\}_{t=1}^n$, with $Z_t = (\tilde{X}_t, \tilde{Y}_t)$. Let

$$Z'_t = (\tilde{X}_t{}', \tilde{Y}_t{}'), \quad t = 1, \ldots, n,$$

be drawn i.i.d. from $P_{\mathrm{mix}}$ (the ghost sample). Then,

$$D_{P_n, \rho}^{mix}(f_*^{mix}) - D_{P_{\mathrm{mix}}, \rho}(f_*^{mix})$$

$$\leq \sup_{f \in \mathcal{NN}_{U,L}^{a_1, a_2}} \mathbb{E}_\sigma \left\{ \mathbb{E}_{P_n, \rho}\big[\tilde{\ell}(f, Z_t)\big] - \mathbb{E}_P\big[\tilde{\ell}(f, Z_t)\big] \right\}$$

$$= \sup_{f \in \mathcal{NN}_{U,L}^{a_1, a_2}} \mathbb{E}_\sigma \left\{ \frac{1}{n} \sum_{t=1}^n \tilde{\ell}(f, Z_t) - \mathbb{E}_{Z'_{1:n}}\left[\frac{1}{n} \sum_{t=1}^n \tilde{\ell}(f, Z'_t)\right] \right\}$$

$$\leq \mathbb{E}_{Z'_{1:n}} \mathbb{E}_\sigma \left\{ \sup_{f \in \mathcal{NN}_{U,L}^{a_1, a_2}} \left[\frac{1}{n} \sum_{t=1}^n \tilde{\ell}(f, Z_t) - \tilde{\ell}(f, Z'_t)\right] \right\}$$

$$= \mathbb{E}_\sigma \left\{ \sup_{f \in \mathcal{NN}_{U,L}^{a_1, a_2}} \frac{1}{n} \sum_{t=1}^n \sigma_t \left[\tilde{\ell}(f, Z_t) - \tilde{\ell}(f, Z'_t)\right] \right\}$$

$$= \mathcal{R}_n(\mathcal{L}^\alpha).$$

Therefore, we got the $B_5 \leq \mathcal{R}_n(\mathcal{L}^\alpha)$. Because for any $f, \tilde{f} \in \mathcal{H}^\alpha$ satisfying $\|f - \tilde{f}\|_\infty \leq u/\mathrm{Lip}^1(\ell)$, it follows

$$\left| \tilde{\ell}(f, Z_t) - \tilde{\ell}(\tilde{f}, Z_t) \right| = \left| \sup_{\tilde{X}_t{}' \in B_\epsilon(\tilde{X}_t)} \ell(f(\tilde{X}_t{}'), \tilde{Y}_t) - \sup_{\tilde{X}_t{}' \in B_\epsilon(\tilde{X}_t)} \ell(\tilde{f}(\tilde{X}_t{}'), \tilde{Y}_t) \right|$$

$$\leq \mathrm{Lip}^1(\ell) \sup_{\tilde{X}_t{}' \in B_\epsilon(\tilde{X}_t)} \left| f(\tilde{X}_t{}') - \tilde{f}(\tilde{X}_t{}') \right| \leq u.$$

Hence, we have

$$\log \mathcal{N}(u, \mathcal{L}^\alpha, \|\cdot\|_\infty) \leq \log \mathcal{N}\left(\frac{u}{\mathrm{Lip}^1(\ell)}, \mathcal{H}^\alpha, \|\cdot\|_\infty\right).$$

Based on the Kolmogorov–Tikhomirov result (1959),

$$\log \mathcal{N}\left(u, \mathcal{L}^\alpha, \|\cdot\|_\infty\right) \lesssim u^{-d/\alpha}.$$

Let $L_2(P_n)$ denote the $L_2$ metric induced by the samples. Then,

$$\log \mathcal{N}\left(u, \mathcal{L}^\alpha, L_2(P_n)\right) \leq \log \mathcal{N}\left(u, \mathcal{L}^\alpha, \|\cdot\|_\infty\right) \lesssim u^{-d/\alpha}.$$

Since $\sup_{f \in \mathcal{H}^\alpha} \|f\|_\infty \leq 1$, there exists a constant $B$ such that

$$\sup_{Z_t \in P_{\mathrm{mix}}} |\tilde{\ell}(f, Z_t)| \leq B \quad \text{for any } f \in \mathcal{H}^\alpha.$$

Combining this with Lemma B.2 and the arguments in (Liu et al., 2024) (Section C.2.3), we get:

$$\hat{\mathcal{R}}_n(\mathcal{L}^\alpha) \lesssim \inf_{\delta \geq 0} \left\{ 4\delta + 12 \int_\delta^1 \sqrt{\frac{\log \mathcal{N}(u, \mathcal{L}^\alpha, \|\cdot\|_\infty)}{n}} \, du \right\}$$

$$\lesssim \inf_{\delta \geq 0} \left\{ \delta + n^{-1/2} \int_\delta^1 u^{-d/(2\alpha)} \, du \right\}$$

$$\lesssim n^{-\min\{1/2, \alpha/d\}} \log^{c(\alpha,d)} n.$$

Therefore,

$$B_5 = D^{mix}_{P_n,\rho}(f^{mix}_*) - D_{P_{\mathrm{mix}},\rho}(f^{mix}_*) \lesssim n^{-\min\{1/2, \alpha/d\}} \log^{c(\alpha,d)} n.$$

### C.1.4 BOUND FOR $B_1$

Define the class of functions $\mathcal{L}_n$ by

$$\mathcal{L}_n = \left\{ \tilde{\ell}(f, z) : \mathcal{Z} \mapsto \mathbb{R} \mid f \in \mathcal{NN}^{a_1,a_2}_{U,L} \right\}.$$

For a given set of samples $z_1, \ldots, z_n$ from $\mathcal{Z}$, the empirical Rademacher complexity of class $\mathcal{L}_n$ is defined by

$$\widehat{\mathcal{R}}_n(\mathcal{L}_n) = \mathbb{E}_\sigma \left\{ \sup_{f \in \mathcal{NN}^{a_1,a_2}_{U,L}} \frac{1}{n} \sum_{i=1}^n \sigma_i \tilde{\ell}(f, z_i) \right\}.$$

As the similar logic for $B_5$, we could see that

$$|D_{P_{\mathrm{mix}},\rho}(\hat{f}^{mix}_{NN}) - D^{mix}_{P_n,\rho}(\hat{f}^{mix}_{NN})| \leq \widehat{\mathcal{R}}_n(\mathcal{L}_n)$$

where $\hat{f}^{mix}_{NN} \in \mathcal{NN}^{a_1,a_2}_{U,L}$. For a given $\tau \in (0, \rho)$, let $C_{B_\rho}(\tau)$ be a $(\tau, \|\cdot\|_\infty)$-cover of $B_\rho(0)$ with the smallest cardinality $M_\tau$. Denote the elements of $C_{B_\rho}(\tau)$ by $\delta_1, \ldots, \delta_{M_\tau}$. It follows by Lemma B.6 that

$$\log M_\tau \leq cd \log(\rho \tau^{-1})$$

for a constant $c$. For any $Z_t = (\tilde{X}_t, \tilde{Y}_t) \in P_{\mathrm{mix}}$, the continuity of $\ell$ and $f$ imply that there exists $\delta' \in B_\rho(0)$ such that

$$\tilde{\ell}(f, Z_t) = \ell(f(\tilde{X}_t + \delta'), \tilde{Y}_t) - \max_k \ell(f(\tilde{X}_t + \delta_k), \tilde{Y}_t)$$

$$\leq \min_k |\ell(f(\tilde{X}_t + \delta'), \tilde{Y}_t) - \ell(f(\tilde{X}_t + \delta_k), \tilde{Y}_t)|$$

$$\leq \mathrm{Lip}^1(\ell) \mathrm{Lip}(f) \|\delta' - \delta_k\|_\infty$$

$$\leq \mathrm{Lip}^1(\ell) \mathrm{Lip}(f) \tau.$$

Therefore, for any $f \in \mathcal{NN}^{a_1,a_2}_{U,L}$,

$$\frac{1}{n} \sum_{t=1}^n \sigma_t \tilde{\ell}(f, Z_t) = \frac{1}{n} \sum_{t=1}^n \left[ \sigma_t \tilde{\ell}(f, Z_t) - \sigma_t \max_k \ell(f(\tilde{X}_t + \delta_k), \tilde{Y}_t) + \sigma_t \max_k \ell(f(\tilde{X}_t + \delta_k), \tilde{Y}_t) \right]$$

$$\leq \mathrm{Lip}^1(\ell) \cdot \frac{a_1 \tau}{2} + \frac{1}{n} \sum_{t=1}^n \left[ \sigma_t \max_k \ell(f(\tilde{X}_t + \delta_k), \tilde{Y}_t) \right]$$

This leads to an upper bound of $\widehat{\mathcal{R}}_n(\mathcal{L}_n)$ as follows:

$$\widehat{\mathcal{R}}_n(\mathcal{L}_n) \leq \mathbb{E}_\sigma \left[ \sup_{f \in \mathcal{NN}^{a_1,a_2}_{U,L}} \frac{1}{n} \sum_{t=1}^n \sigma_t \max_k \ell(f(\tilde{X}_t + \delta_k), \tilde{Y}_t) \right] + \mathrm{Lip}^1(\ell) \cdot \frac{a_1 \tau}{2}$$

Define the class

$$\mathcal{L}_{n,\tau} = \left\{ \max_k \ell(f(x + \delta_k), y) : \mathcal{Z} \mapsto \mathbb{R} \mid f \in \mathcal{NN}_{U,L}^{a_1, a_2} \right\}.$$

Let $\mathcal{N}(u, \mathcal{L}_{n,\tau}, L_\infty(P_n))$ denote the covering number of the class $\mathcal{L}_{n,\tau}$ under the data-dependent $L_\infty$ metric. Define

$$S_{n,M_\tau} = \{x_i + \delta_k : i = 1, \ldots, n, \ K = 1, \ldots, M_\tau\},$$

and let $\mathcal{N}(u, \mathcal{NN}_{U,L}^{a_1, a_2}, L_\infty(P_{nM_\tau}))$ denote the covering number of the class $\mathcal{NN}_{U,L}^{a_1, a_2}$ under the $L_\infty$ metric on the set $S_{n,M_\tau}$. For any $f, f' \in \mathcal{NN}_{U,L}^{a_1, a_2}$, if $\max_{t,k} |f(\tilde{X}_t + \delta_k) - f'(\tilde{X}_t + \delta_k)| \le u$, then

$$\max_t \left| \max_k \ell(f(\tilde{X}_t + \delta_k), \tilde{Y}_t) - \max_k \ell(f'(\tilde{X}_t + \delta_k), \tilde{Y}_t) \right|$$

$$\le \max_{t,k} \left| \ell(f(\tilde{X}_t + \delta_k), \tilde{Y}_t) - \ell(f'(\tilde{X}_t + \delta_k), \tilde{Y}_t) \right|$$

$$\le \mathrm{Lip}^1(\ell) \cdot u$$

Hence,

$$\mathcal{N}(u, \mathcal{L}_{n,\tau}, L_\infty(P_n)) \le \mathcal{N}(u/\mathrm{Lip}^1(\ell), \mathcal{NN}_{U,L}^{a_1, a_2}, L_\infty(P_{nM_\tau})).$$

Suppose the functions in $\mathcal{NN}_{U,L}^{a_1, a_2}$ are uniformly bounded. Then the uniform covering number of $\mathcal{NN}_{U,L}^{a_1, a_2}$ is defined by

$$\mathcal{N}_\infty(u, \mathcal{NN}_{U,L}^{a_1, a_2}, nM_\tau) = \sup_{P_n} \mathcal{N}(u, \mathcal{NN}_{U,L}^{a_1, a_2}, L_\infty(P_n M_\tau)),$$

where the supremum runs over all data sets of size $n$. Combining Lemma B.5 and Lemma B.4, we derive

$$\log \mathcal{N}(u, \mathcal{L}_{n,\tau}, L_\infty(P_n)) \le \log \mathcal{N}_\infty(u, \mathcal{NN}_{U,L}^{a_1, a_2}, nM_\tau)$$

$$\le B_1 W^2 L^2 (L + \log(W^2 L)) \log(u^{-1} n M_\tau),$$

for a constant $B_1$. Since $\mathcal{NN}_{U,L}^{a_1, a_2}$ is bounded and $\ell$ is continuous, there exists $B_2 > 0$ such that

$$\sup_{Z_t \in P_\mathrm{mix}} \max_k \ell(f(\tilde{X}_t + \delta_k), \tilde{Y}_t) \le B_2 \quad \text{for any } f \in \mathcal{NN}_{U,L}^{a_1, a_2}.$$

Combining above with Lemma B.2, we have

$$\mathbb{E}_\sigma \left\{ \sup_{f \in \mathcal{NN}_{U,L}^{a_1, a_2}} \frac{1}{n} \sum_{t=1}^n \sigma_t \max_k \ell(f(\tilde{X}_t + \delta_k), \tilde{Y}_t) \right\}$$

$$\le \inf_{\delta \ge 0} \left\{ 4\delta + 12 \int_\delta^{B_2} \sqrt{\frac{\log \mathcal{N}(u, \mathcal{L}_{n,\tau}, L_\infty(P_n))}{n}} \, du \right\}.$$

Hence,

$$\lesssim \inf_{\delta \ge 0} \left[ \delta + \left( \sqrt{UL^2 + UL \log(U)} \right) n^{-1/2} \cdot \int_\delta^{B_2} \left[ \sqrt{\log(u^{-1})} + \sqrt{\log n} + \sqrt{\log M_\tau} \right] du \right].$$

Thus,

$$\lesssim \left( \sqrt{UL^2 + UL \log(U)} \right) n^{-1/2} \left\{ \sqrt{\log n} + \sqrt{\log(\rho \tau^{-1})} \right\}.$$

Therefore, by selecting $\tau$ such that $\rho \tau^{-1} = \mathcal{O}(n)$, we show

$$|D_{P_\mathrm{mix}, \rho}(\hat{f}_{NN}^{mix}) - D_{P_n, \rho}^{mix}(\hat{f}_{NN}^{mix})| \lesssim \rho n^{-1} + \left( \sqrt{U + U \log(U)} \right) n^{-1/2} \sqrt{\log n}.$$

Now if we summarize them all, we get:

$$|D_{P_\mathrm{mix}, \rho}(\hat{f}_{NN}^{mix}) - D_{P_\mathrm{mix}, \rho}(f_*^{mix})| \lesssim \frac{\alpha^d}{Z^\alpha} + \rho \sqrt{\frac{t}{2n}} + \rho^2 + n^{-\min\{1/2, \alpha/d\}} \log^{c(\alpha, d)} n + \rho n^{-1}$$

$$+ \left( \sqrt{U + U \log(U)} \right) n^{-1/2} \sqrt{\log n}.$$

# D    ADDITIONAL RESULTS

Table 4: Performance comparison under severity level 5 on CIFAR-10-C (higher values are better).

| Method | Noise | | | Blur | | | | Weather | | | | Digital | | | |
|---|---|---|---|---|---|---|---|---|---|---|---|---|---|---|---|
| | White | Shot | Impulse | Defocus | Glass | Motion | Zoom | Snow | Frost | Fog | Bright | Contrast | Elastic | Pixel | JPEG |
| Baseline | 24.90 | 32.44 | 25.03 | 57.08 | 49.45 | 68.46 | 64.28 | 76.98 | 65.38 | 71.77 | 91.28 | 36.20 | 75.56 | 45.85 | 72.05 |
| Mixup | 32.56 | 39.17 | 21.00 | 69.49 | 66.42 | 73.00 | 71.15 | 83.36 | **82.78** | **81.18** | **90.96** | **65.14** | 80.34 | 61.30 | 76.61 |
| Mixup + DRO | **44.93** | **51.86** | **31.43** | **72.17** | **69.79** | **75.38** | **75.83** | 83.82 | 82.03 | 80.74 | 89.44 | 61.15 | **83.57** | **65.79** | **80.24** |
| AugMix | **65.98** | 69.03 | 59.01 | 87.16 | **67.54** | 86.77 | 87.45 | 87.53 | 85.04 | **88.92** | 94.06 | 89.29 | **78.97** | 55.52 | 79.22 |
| AugMix + DRO | 65.89 | **69.54** | **66.55** | **89.50** | 67.16 | **87.39** | **88.92** | **87.81** | **86.33** | 87.56 | **94.69** | **90.74** | 78.28 | **65.57** | **80.88** |
| NoisyMix | 87.17 | 88.59 | 94.60 | 90.34 | 82.69 | 88.08 | 90.79 | 87.81 | 88.74 | 79.16 | 92.58 | 64.03 | 85.83 | 77.42 | 88.63 |
| NoisyMix + DRO | **87.98** | **89.20** | **94.82** | **91.08** | **83.77** | **89.35** | **91.37** | **88.49** | **89.55** | **81.65** | **93.19** | **67.95** | **86.58** | **77.71** | **88.66** |

Table 5: Performance comparison under severity level 5 on CIFAR-100-C (higher values are better).

| Method | Noise | | | Blur | | | | Weather | | | | Digital | | | |
|---|---|---|---|---|---|---|---|---|---|---|---|---|---|---|---|
| | White | Shot | Impulse | Defocus | Glass | Motion | Zoom | Snow | Frost | Fog | Bright | Contrast | Elastic | Pixel | JPEG |
| Baseline | 10.16 | 11.67 | 6.31 | 32.61 | 18.96 | 41.97 | 39.91 | 44.22 | 33.00 | 37.93 | 65.48 | 17.87 | 47.41 | 23.72 | 40.39 |
| Mixup | **11.68** | **15.03** | **5.33** | 39.80 | 24.54 | 48.64 | 45.75 | 50.35 | 41.91 | 48.56 | 65.85 | 37.59 | **52.01** | **30.09** | **46.53** |
| Mixup + DRO | 10.02 | 13.11 | 4.05 | **42.09** | **24.97** | **51.68** | **47.78** | **53.78** | **44.07** | **52.15** | **66.30** | **41.20** | 48.71 | 29.08 | 45.51 |
| AugMix | 36.99 | 39.93 | 37.06 | 60.33 | 22.36 | 59.17 | 61.30 | 57.07 | 47.57 | **60.52** | 71.52 | 62.57 | 46.70 | **31.70** | 49.66 |
| AugMix + DRO | **40.80** | **43.30** | **40.94** | **63.51** | **27.67** | **60.97** | **63.19** | **58.84** | **51.33** | 58.31 | **72.08** | **65.19** | **47.26** | 30.71 | **50.19** |
| NoisyMix | 60.21 | 62.42 | 77.68 | **71.59** | 54.96 | 69.18 | 71.31 | 63.94 | 62.45 | **54.48** | 70.82 | 40.14 | **64.31** | 57.37 | 64.68 |
| NoisyMix + DRO | **63.30** | **65.03** | **78.34** | 71.57 | **55.77** | **69.30** | **71.73** | **65.27** | **63.62** | 53.62 | **71.13** | **40.98** | 64.09 | **57.98** | **65.51** |

Table 6: Performance comparison under different corruption types on CIFAR-10-C (higher values are better).

| Method | Noise | | | Blur | | | | Weather | | | | Digital | | | |
|---|---|---|---|---|---|---|---|---|---|---|---|---|---|---|---|
| | White | Shot | Impulse | Defocus | Glass | Motion | Zoom | Snow | Frost | Fog | Bright | Contrast | Elastic | Pixel | JPEG |
| Baseline | 44.45 | 57.31 | 56.55 | 83.42 | 54.90 | 79.74 | 79.28 | 82.89 | 78.66 | 88.38 | 93.85 | 77.42 | 84.91 | 74.91 | 79.17 |
| Mixup | 54.17 | 64.81 | 53.16 | 87.11 | 69.50 | 82.00 | 82.66 | **88.01** | **88.17** | **91.63** | **93.84** | **87.18** | 86.95 | 83.16 | 82.95 |
| Mixup + DRO | **61.67** | **70.88** | **56.42** | **87.36** | **73.35** | **83.21** | **84.80** | 87.59 | 87.46 | 90.33 | 92.64 | 84.88 | **88.03** | **84.55** | **84.71** |
| AugMix | 77.68 | 82.63 | 82.03 | 92.85 | **73.27** | 90.45 | 91.50 | 90.02 | 89.84 | 93.72 | 95.11 | 93.75 | 88.70 | 80.40 | 84.45 |
| AugMix + DRO | **78.20** | **83.33** | **85.28** | **93.87** | 73.25 | **91.05** | **92.47** | **91.03** | **91.00** | **93.84** | **95.57** | **94.39** | **89.15** | **85.10** | **85.80** |
| NoisyMix | 90.53 | 91.98 | 95.11 | 93.63 | 86.08 | 91.40 | 92.73 | 91.25 | 91.63 | 90.59 | 94.55 | 87.02 | 90.92 | 89.14 | 90.82 |
| NoisyMix + DRO | **90.95** | **92.42** | **95.33** | **94.12** | **86.87** | **92.20** | **93.27** | **91.55** | **92.16** | **91.61** | **94.93** | **88.28** | **91.47** | **89.33** | **90.88** |

Table 7: Performance comparison under different corruption types on CIFAR-100-C (higher values are better).

| Method | Noise | | | Blur | | | | Weather | | | | Digital | | | |
|---|---|---|---|---|---|---|---|---|---|---|---|---|---|---|---|
| | White | Shot | Impulse | Defocus | Glass | Motion | Zoom | Snow | Frost | Fog | Bright | Contrast | Elastic | Pixel | JPEG |
| Baseline | 21.62 | 29.69 | 24.90 | 60.16 | 20.77 | 53.82 | 53.77 | 54.04 | 48.13 | 63.44 | 73.21 | 54.07 | 60.15 | 51.13 | 49.33 |
| Mixup | **27.87** | **36.86** | 24.12 | 63.45 | 26.14 | 59.13 | 58.08 | 59.78 | 54.61 | 67.80 | 73.11 | 63.91 | 62.83 | 57.53 | 54.09 |
| Mixup + DRO | 26.17 | 35.94 | **26.26** | **65.97** | **26.56** | **62.01** | **60.58** | **62.87** | **57.26** | **70.51** | **74.97** | **66.31** | **63.53** | **58.11** | **54.14** |
| AugMix | 49.57 | 55.47 | 60.16 | 71.24 | 27.35 | 66.05 | 67.97 | 64.76 | 59.12 | **72.17** | 75.43 | 72.22 | 63.62 | **57.67** | 56.32 |
| AugMix + DRO | **52.43** | **57.60** | **61.55** | **72.71** | **33.41** | **67.33** | **69.66** | **66.16** | **61.58** | 71.91 | **75.75** | **73.72** | **63.99** | 57.48 | **56.79** |
| NoisyMix | 67.12 | 70.32 | 78.44 | 76.27 | 60.32 | 73.34 | 74.69 | 70.30 | 69.27 | **70.59** | 76.16 | 66.82 | **72.13** | 70.52 | 68.93 |
| NoisyMix + DRO | **68.93** | **71.78** | **79.01** | **76.63** | **60.94** | **73.50** | **75.03** | **71.26** | **70.01** | 70.41 | **76.40** | **67.14** | 72.01 | **70.92** | **69.49** |

Table 8: Performance comparison under different corruption types on Tiny ImageNet-C (higher values are better).

| Method | Noise | | | Blur | | | | Weather | | | | Digital | | | |
|---|---|---|---|---|---|---|---|---|---|---|---|---|---|---|---|
| | Gaussian | Shot | Impulse | Defocus | Glass | Motion | Zoom | Snow | Frost | Fog | Bright | Contrast | Elastic | Pixel | JPEG |
| Baseline | 12.25 | 15.24 | 13.86 | 11.50 | 11.47 | 15.08 | 13.01 | 17.42 | 19.04 | 19.07 | 24.20 | 8.51 | 18.78 | 25.58 | 24.62 |
| Mixup | 20.81 | 23.64 | 14.90 | 14.71 | 14.84 | 18.66 | 16.06 | **24.52** | 26.43 | 23.78 | 27.81 | 15.58 | 21.73 | 22.80 | **29.92** |
| Mixup + DRO | **23.32** | **26.05** | **17.28** | **16.97** | **17.19** | **21.13** | **18.45** | 23.42 | **27.33** | **26.83** | **28.75** | **16.56** | **22.68** | **26.78** | 27.87 |
| AugMix | 29.42 | 32.35 | 20.47 | **22.74** | 17.46 | 23.04 | 20.59 | 31.63 | 30.81 | 29.99 | 34.36 | **18.82** | 28.37 | 26.61 | 27.94 |
| AugMix + DRO | **31.81** | **34.68** | **22.71** | 22.02 | **19.77** | **25.33** | **22.85** | **32.61** | **34.81** | **30.93** | **35.25** | 17.78 | **29.30** | **29.59** | **28.90** |
| NoisyMix | 25.33 | 27.78 | 23.95 | 22.30 | 16.99 | 25.92 | 23.42 | 32.12 | 34.85 | 36.17 | 39.92 | 21.13 | **32.79** | 29.64 | 33.74 |
| NoisyMix + DRO | **26.75** | **30.17** | **25.33** | **23.63** | **18.40** | **27.43** | **24.82** | **32.35** | **35.05** | **36.41** | **40.12** | **21.34** | 31.98 | **29.88** | **33.95** |

Table 9: Performance comparison under severity level 5 on Tiny ImageNet-C (higher values are better).

| Method | Noise | | | Blur | | | | Weather | | | | Digital | | | |
|---|---|---|---|---|---|---|---|---|---|---|---|---|---|---|---|
| | Gaussian | Shot | Impulse | Defocus | Glass | Motion | Zoom | Snow | Frost | Fog | Bright | Contrast | Elastic | Pixel | JPEG |
| Baseline | 3.83 | 4.90 | 3.00 | 4.47 | 4.08 | 10.02 | 9.12 | 13.20 | 13.34 | 8.99 | 16.43 | 1.55 | 12.36 | 16.52 | 18.72 |
| Mixup | 10.12 | **13.64** | 4.87 | 7.62 | 5.55 | 13.41 | 10.82 | **17.97** | **25.85** | 14.61 | 16.51 | 2.43 | 14.46 | 14.24 | **22.87** |
| Mixup + DRO | **14.09** | 12.37 | **7.55** | **8.85** | **8.05** | **16.42** | **14.40** | 16.69 | 24.58 | **19.40** | **21.35** | **4.22** | **18.23** | **17.91** | 21.66 |
| AugMix | 19.24 | 20.88 | 8.35 | **13.33** | 6.09 | 15.65 | 14.40 | 26.04 | 25.18 | 17.00 | 23.04 | **5.71** | **20.39** | 12.44 | **20.43** |
| AugMix + DRO | **21.16** | **24.35** | **10.93** | 11.64 | **8.64** | **18.34** | **18.36** | **29.67** | **30.03** | **20.75** | **25.77** | 4.59 | 19.22 | **16.31** | 19.28 |
| NoisyMix | **19.90** | 18.60 | 11.06 | 7.92 | **11.60** | 18.67 | 14.48 | 23.43 | **29.59** | 19.51 | 25.48 | **6.15** | 19.49 | 10.84 | 21.93 |
| NoisyMix + DRO | 18.10 | **19.78** | **12.17** | **11.93** | 9.83 | **19.26** | **16.64** | **26.64** | 27.78 | **24.76** | **30.65** | 4.80 | **21.67** | **12.08** | **26.62** |

Table 10: Average accuracy (%) of Preact-ResNet-18 on standard and "-C" corruption benchmarks.

| Method | CIFAR-10 | | CIFAR-100 | | Tiny-ImageNet | |
|---|---|---|---|---|---|---|
| | Clean ↑ | -C ↑ | Clean ↑ | -C ↑ | Clean ↑ | -C ↑ |
| Baseline | 95.21 | 74.39 | 77.52 | 47.88 | 45.20 | 16.64 |
| Mixup | **95.44** | 79.69 | 79.65 | 52.62 | 47.82 | 21.08 |
| Mixup + DRO | 94.97 | **81.19** | **79.89** | **54.08** | **48.50** | **22.74** |
| AugMix | 95.55 | 87.09 | 77.42 | 61.28 | 54.80 | 26.16 |
| AugMix + DRO | **96.07** | **88.22** | **77.83** | **62.82** | **54.97** | **27.89** |
| NoisyMix | **95.37** | 91.15 | 79.62 | 71.01 | 57.93 | 28.04 |
| NoisyMix + DRO | 95.22 | **91.69** | **79.77** | **71.56** | **57.96** | **29.11** |

Table 11: Adversarial robustness accuracy comparison of ConvNeXt-S under multiple attacks with different $\epsilon$ values in Fashion-MNIST-$\epsilon$ datasets.

| Method | AA(%) | | | PGD(%) | | | C&W(%) | | | FAB-T(%) | Square(%) |
|---|---|---|---|---|---|---|---|---|---|---|---|
| | 2/255 | 4/255 | 8/255 | 4/255 | 8/255 | 16/255 | $c=1$ | $c=5$ | $c=10$ | 8/255 | 8/255 |
| Baseline | 32.00 | 27.85 | 20.65 | 28.81 | 21.56 | 12.82 | 22.58 | 22.14 | 22.11 | 89.90 | 24.05 |
| Mixup | 54.20 | 51.40 | 45.80 | 53.12 | 48.02 | 39.27 | 48.69 | 47.88 | 47.82 | 89.75 | 48.60 |
| Mixup + DRO | 61.35 | 59.50 | 52.75 | 61.18 | 56.41 | 45.12 | 55.68 | 54.24 | 54.06 | 90.05 | 56.45 |
| AugMix | 67.10 | 64.35 | 57.10 | 65.06 | 57.99 | 46.59 | 58.69 | 57.51 | 57.48 | 90.15 | 61.30 |
| AugMix + DRO | 72.55 | 69.30 | 65.65 | 71.18 | 64.86 | 58.48 | 67.54 | 65.32 | 65.24 | 90.40 | 68.40 |
| NoisyMix | 69.65 | 67.70 | 64.70 | 69.64 | 66.09 | 59.27 | 66.36 | 64.78 | 64.66 | 85.80 | 66.35 |
| NoisyMix + DRO | 75.80 | 73.25 | 70.15 | 74.68 | 69.36 | 62.84 | 73.62 | 71.34 | 71.16 | 86.20 | 72.60 |
| TRADES | 62.20 | 61.35 | 59.35 | 63.63 | 61.56 | 58.03 | 60.97 | 59.91 | 59.79 | 87.85 | 60.30 |
| DRO-AT | 51.20 | 50.95 | 50.25 | 50.63 | 50.15 | 49.36 | 50.33 | 50.02 | 49.95 | 87.70 | 50.45 |
| MART | 46.95 | 46.60 | 45.60 | 47.82 | 46.73 | 44.94 | 47.04 | 46.28 | 46.15 | 87.50 | 45.85 |

Table 12: Average accuracy (%) of ConvNeXt-S on standard and "-C" corruption benchmarks.

| Method | CIFAR-10 ($\uparrow$%) | CIFAR-10-C ($\uparrow$%) | CIFAR-100 ($\uparrow$%) | CIFAR-100-C ($\uparrow$%) |
|---|---|---|---|---|
| Baseline | 62.99 | 51.10 | 43.68 | 33.05 |
| Mixup | 59.21 | 50.55 | 31.44 | 24.77 |
| Mixup + DRO | 59.43 | 51.74 | 31.29 | 25.91 |
| AugMix | 65.12 | 57.93 | 44.09 | 37.77 |
| AugMix + DRO | 65.07 | 59.49 | 44.37 | 39.13 |
| NoisyMix | 49.34 | 45.38 | 24.14 | 21.52 |
| NoisyMix + DRO | 49.56 | 45.97 | 24.28 | 22.16 |

Table 13: Average time and space cost (%) of ConvNeXt-S on Fashion-MNIST.

| Method | Time(s) | GPU Memory(MiB) |
|---|---|---|
| Baseline | $34.25 \pm 1.15$ | 889 |
| Mixup | $34.61 \pm 1.38$ | 891 |
| Mixup + DRO | $52.22 \pm 1.27$ | 1597 |
| AugMix | $49.85 \pm 1.48$ | 1730 |
| AugMix + DRO | $76.27 \pm 1.51$ | 3072 |
| NoisyMix | $64.02 \pm 1.57$ | 1742 |
| NoisyMix + DRO | $95.39 \pm 1.63$ | 3105 |
| TRADES | $303.75 \pm 14.84$ | 2178 |
| DRO-AT | $222.17 \pm 16.51$ | 902 |
| MART | $241.26 \pm 9.45$ | 1334 |

Table 14: Sensitivity study on CIFAR-100 dataset with different $\rho$ values.

| $\rho$ | CIFAR-100 (%) | CIFAR-100-C (%) |
|---|---|---|
| $1.00 * 10^{-3}$ | 77.65 | 61.89 |
| $1.05 * 10^{-3}$ | 77.31 | 61.62 |
| $1.10 * 10^{-3}$ | 76.81 | 61.97 |
| $1.15 * 10^{-3}$ | 77.69 | 62.33 |
| $1.20 * 10^{-3}$ | 77.83 | 62.82 |
| $1.25 * 10^{-3}$ | 77.24 | 61.75 |

# E    REFINED CIFAR-C

We aim to address the issue with the severity settings in the CIFAR-C datasets. In the original CIFAR-10-C dataset, model performance varies significantly across different corruption types, even at the same severity level. As the severity increases, this performance gap widens, with differences in accuracy exceeding 40% in extreme cases. Additionally, at severity level 1—the lowest level—most baseline models, such as ResNet-18, already achieve near-maximum accuracy, around 95%. This leaves minimal room for observing improvements, rendering severity level 1 practically uninformative.

To address this, we propose redefining the severity levels. Given that the predominant architectures used in image classification today are Transformer-based and ResNet-based models, we first designed our approach with ResNet performance in mind. We set the baseline accuracy at 50%, corresponding to the probability of random guessing in a binary decision task (e.g., 'is this class A or not?'). For the upper bound, we set the initial accuracy at approximately 85%, acknowledging that 95% represents a practical ceiling for CIFAR-10-C. This adjustment provides sufficient room to observe robustness improvements across a broader range of models.

Table 15 and 16 show that, in CIFAR-100-C, ResNet performance generally drops by approximately 30% compared to CIFAR-10-C. To account for this, we adjust our standard accuracy range downward by 30%. Specifically, we redefine the severity levels such that at severity 5, the accuracy is set at 60%, decreasing in increments of 10% down to 20% at severity 1. This proposed configuration ensures more consistent accuracy across different corruption types at the same severity level for ResNet architectures of varying depths, allowing for a more reliable evaluation of model robustness. The detailed performance of ResNet-18, ResNet-34, ResNet-50, and ResNet-101 on the refined CIFAR-10-C and CIFAR-100-C benchmarks can be found in table 17, 18,19,20, 21, 22, 23, 24.

Table 15: Performance on Refined CIFAR–10–C (% accuracy)

| Model | S1 | S2 | S3 | S4 | S5 |
|---|---|---|---|---|---|
| ResNet-18 | 86.04 | 79.03 | 70.00 | 61.83 | 51.59 |
| ResNet-34 | 86.38 | 79.47 | 70.60 | 62.95 | 52.95 |
| ResNet-50 | 88.10 | 80.18 | 70.18 | 61.69 | 50.68 |
| ResNet-101 | 87.60 | 79.54 | 69.51 | 60.85 | 49.74 |

Table 16: Performance on Refined CIFAR–100–C (% accuracy)

| Model | S1 | S2 | S3 | S4 | S5 |
|---|---|---|---|---|---|
| ResNet-18 | 59.93 | 50.78 | 40.85 | 30.77 | 20.53 |
| ResNet-34 | 57.87 | 49.91 | 41.04 | 31.69 | 21.95 |
| ResNet-50 | 65.11 | 55.83 | 45.51 | 35.05 | 23.86 |
| ResNet-101 | 63.80 | 54.58 | 44.65 | 34.53 | 23.87 |

Table 17: Accuracy of ResNet-18 on Refined CIFAR-10-C under different severity levels

| Type | S1 | S2 | S3 | S4 | S5 |
|------|------|------|------|------|------|
| gaussian | 86.20 | 78.01 | 69.30 | 61.61 | 51.22 |
| shot | 86.51 | 77.16 | 69.00 | 60.36 | 50.88 |
| impulse | 85.01 | 78.65 | 69.19 | 59.73 | 50.82 |
| defocus | 87.03 | 79.53 | 69.27 | 60.50 | 46.63 |
| glass | 84.88 | 79.16 | 67.59 | 60.73 | 52.56 |
| motion | 85.30 | 78.56 | 70.38 | 60.05 | 52.34 |
| zoom | 85.81 | 79.94 | 71.49 | 64.12 | 54.39 |
| snow | 87.02 | 78.12 | 69.72 | 64.15 | 54.69 |
| frost | 85.86 | 77.90 | 69.68 | 63.13 | 54.17 |
| fog | 85.38 | 78.62 | 69.98 | 60.35 | 49.59 |
| bright | 86.83 | 80.06 | 71.70 | 62.78 | 52.40 |
| contrast | 86.62 | 81.31 | 71.98 | 63.79 | 52.49 |
| elastic | 85.55 | 78.86 | 69.09 | 59.94 | 51.29 |
| pixelate | 86.49 | 80.47 | 71.57 | 64.92 | 53.53 |
| jpeg | 86.18 | 79.03 | 70.01 | 61.33 | 46.79 |
| **Avg** | **86.04** | **79.03** | **70.00** | **61.83** | **51.59** |

Table 18: Accuracy of ResNet-34 on Refined CIFAR-10-C under different severity levels

| Type | S1 | S2 | S3 | S4 | S5 |
|------|------|------|------|------|------|
| gaussian | 86.20 | 78.01 | 69.30 | 61.61 | 51.22 |
| shot | 86.51 | 77.16 | 69.00 | 60.36 | 50.88 |
| impulse | 85.01 | 78.65 | 69.19 | 59.73 | 50.82 |
| defocus | 87.03 | 79.53 | 69.27 | 60.50 | 46.63 |
| glass | 84.88 | 79.16 | 67.59 | 60.73 | 52.56 |
| motion | 85.30 | 78.56 | 70.38 | 60.05 | 52.34 |
| zoom | 85.81 | 79.94 | 71.49 | 64.12 | 54.39 |
| snow | 87.02 | 78.12 | 69.72 | 64.15 | 54.69 |
| frost | 85.86 | 77.90 | 69.68 | 63.13 | 54.17 |
| fog | 85.38 | 78.62 | 69.98 | 60.35 | 49.59 |
| bright | 86.83 | 80.06 | 71.70 | 62.78 | 52.40 |
| contrast | 86.62 | 81.31 | 71.98 | 63.79 | 52.49 |
| elastic | 85.55 | 78.86 | 69.09 | 59.94 | 51.29 |
| pixelate | 86.49 | 80.47 | 71.57 | 64.92 | 53.53 |
| jpeg | 86.18 | 79.03 | 70.01 | 61.33 | 46.79 |
| **Avg** | **86.04** | **79.03** | **70.00** | **61.83** | **51.59** |

Table 19: Accuracy of ResNet-50 on Refined CIFAR-10-C under different severity levels

| Type | S1 | S2 | S3 | S4 | S5 |
|------|------|------|------|------|------|
| gaussian | 88.40 | 80.30 | 70.74 | 62.44 | 50.43 |
| shot | 89.17 | 79.85 | 71.92 | 62.72 | 51.95 |
| impulse | 86.97 | 81.18 | 70.95 | 61.20 | 50.78 |
| defocus | 89.28 | 78.65 | 64.31 | 53.83 | 39.72 |
| glass | 86.69 | 80.06 | 67.51 | 61.80 | 52.19 |
| motion | 87.08 | 78.72 | 69.60 | 59.01 | 49.99 |
| zoom | 87.74 | 79.16 | 68.41 | 59.56 | 48.26 |
| snow | 88.91 | 79.66 | 69.56 | 65.10 | 53.84 |
| frost | 88.10 | 79.13 | 72.06 | 63.32 | 54.37 |
| fog | 87.65 | 81.19 | 72.50 | 64.43 | 52.57 |
| bright | 88.92 | 81.53 | 71.93 | 62.97 | 52.53 |
| contrast | 88.85 | 83.61 | 75.32 | 67.76 | 57.22 |
| elastic | 86.89 | 79.88 | 70.57 | 60.93 | 51.27 |
| pixelate | 88.32 | 79.58 | 68.12 | 60.87 | 49.69 |
| jpeg | 88.53 | 80.15 | 69.14 | 59.44 | 45.43 |
| **Avg** | **88.10** | **80.18** | **70.18** | **61.69** | **50.68** |

Table 20: Accuracy of ResNet-101 on Refined CIFAR-10-C under different severity levels

| Type | S1 | S2 | S3 | S4 | S5 |
|------|------|------|------|------|------|
| gaussian | 87.93 | 79.14 | 70.12 | 61.87 | 51.49 |
| shot | 88.16 | 78.44 | 69.57 | 61.25 | 51.38 |
| impulse | 86.71 | 80.30 | 70.53 | 60.33 | 51.44 |
| defocus | 88.89 | 78.76 | 65.34 | 54.08 | 38.82 |
| glass | 86.35 | 79.18 | 67.53 | 63.04 | 51.37 |
| motion | 86.86 | 78.19 | 69.17 | 57.63 | 48.23 |
| zoom | 87.30 | 79.16 | 68.50 | 60.03 | 48.15 |
| snow | 87.98 | 78.10 | 68.25 | 63.24 | 50.69 |
| frost | 87.43 | 77.59 | 68.63 | 60.51 | 51.59 |
| fog | 87.01 | 80.17 | 70.99 | 61.46 | 49.88 |
| bright | 88.28 | 80.47 | 69.92 | 60.25 | 50.06 |
| contrast | 88.22 | 82.30 | 73.23 | 65.31 | 54.19 |
| elastic | 86.73 | 79.79 | 69.99 | 60.35 | 51.00 |
| pixelate | 88.19 | 81.69 | 70.86 | 63.70 | 51.89 |
| jpeg | 87.94 | 79.82 | 70.00 | 59.66 | 45.87 |
| **Avg** | **87.60** | **79.54** | **69.51** | **60.85** | **49.74** |

Table 21: Accuracy of ResNet-18 on Refined CIFAR-100-C under different severity levels

| Type | S1 | S2 | S3 | S4 | S5 |
|------|------|------|------|------|------|
| gaussian | 61.88 | 49.74 | 39.15 | 31.80 | 19.09 |
| shot | 61.45 | 50.45 | 42.01 | 29.19 | 18.49 |
| impulse | 61.29 | 52.34 | 42.23 | 29.00 | 22.49 |
| defocus | 61.01 | 50.95 | 40.08 | 33.01 | 21.55 |
| glass | 59.79 | 50.35 | 38.33 | 25.36 | 19.48 |
| motion | 59.86 | 52.02 | 41.66 | 32.28 | 20.51 |
| zoom | 57.44 | 48.62 | 40.52 | 29.88 | 18.71 |
| snow | 59.39 | 53.55 | 43.97 | 32.79 | 21.91 |
| frost | 58.51 | 48.81 | 42.59 | 30.20 | 19.89 |
| fog | 58.13 | 52.68 | 40.17 | 31.70 | 23.20 |
| bright | 58.87 | 52.06 | 41.42 | 29.47 | 20.36 |
| contrast | 60.28 | 51.64 | 39.66 | 32.77 | 21.74 |
| elastic | 59.93 | 49.45 | 40.41 | 32.73 | 19.08 |
| pixelate | 61.13 | 49.21 | 40.00 | 31.69 | 19.89 |
| jpeg | 59.99 | 49.76 | 40.59 | 29.75 | 21.60 |
| **Avg** | **59.93** | **50.78** | **40.85** | **30.77** | **20.53** |

Table 22: Accuracy of ResNet-34 on Refined CIFAR-100-C under different severity levels

| Type | S1 | S2 | S3 | S4 | S5 |
|------|------|------|------|------|------|
| gaussian | 59.36 | 49.87 | 41.46 | 34.49 | 23.00 |
| shot | 58.79 | 50.88 | 44.08 | 32.15 | 22.51 |
| impulse | 59.12 | 52.33 | 43.06 | 32.07 | 26.45 |
| defocus | 59.26 | 50.70 | 41.72 | 35.83 | 24.92 |
| glass | 58.06 | 50.98 | 42.83 | 31.56 | 25.49 |
| motion | 58.18 | 51.20 | 41.39 | 32.90 | 21.23 |
| zoom | 56.01 | 48.39 | 41.47 | 31.23 | 19.61 |
| snow | 57.54 | 50.65 | 40.63 | 29.72 | 19.87 |
| frost | 56.44 | 45.89 | 40.51 | 29.81 | 19.59 |
| fog | 55.82 | 50.15 | 38.11 | 29.57 | 21.63 |
| bright | 57.36 | 49.94 | 39.84 | 28.46 | 20.72 |
| contrast | 58.18 | 48.98 | 37.74 | 31.28 | 20.80 |
| elastic | 57.92 | 48.92 | 40.13 | 31.88 | 18.77 |
| pixelate | 58.61 | 50.22 | 41.76 | 34.05 | 22.27 |
| jpeg | 57.45 | 49.58 | 40.81 | 30.37 | 22.37 |
| **Avg** | **57.87** | **49.91** | **41.04** | **31.69** | **21.95** |

Table 23: Accuracy of ResNet-50 on Refined CIFAR-100-C under different severity levels

| Type | S1 | S2 | S3 | S4 | S5 |
|------|------|------|------|------|------|
| gaussian | 66.76 | 54.96 | 44.10 | 36.41 | 22.75 |
| shot | 66.23 | 56.02 | 47.55 | 34.17 | 22.47 |
| impulse | 66.29 | 59.35 | 48.46 | 35.49 | 28.00 |
| defocus | 66.02 | 55.38 | 42.41 | 35.78 | 23.08 |
| glass | 65.20 | 53.53 | 40.20 | 27.68 | 21.45 |
| motion | 64.95 | 55.80 | 44.60 | 34.38 | 20.64 |
| zoom | 62.35 | 52.88 | 43.74 | 32.22 | 19.69 |
| snow | 65.11 | 59.60 | 51.18 | 39.26 | 27.21 |
| frost | 64.19 | 54.81 | 48.72 | 36.35 | 25.50 |
| fog | 63.85 | 58.84 | 45.78 | 37.20 | 27.33 |
| bright | 64.84 | 58.08 | 48.39 | 36.03 | 25.60 |
| contrast | 65.36 | 58.06 | 47.06 | 40.63 | 28.80 |
| elastic | 64.71 | 53.68 | 44.66 | 36.46 | 21.55 |
| pixelate | 66.05 | 52.94 | 43.46 | 33.07 | 22.21 |
| jpeg | 64.71 | 53.56 | 42.33 | 30.69 | 21.55 |
| **Avg** | **65.11** | **55.83** | **45.51** | **35.05** | **23.86** |

Table 24: Accuracy of ResNet-101 on Refined CIFAR-100-C under different severity levels

| Type | S1 | S2 | S3 | S4 | S5 |
|------|------|------|------|------|------|
| gaussian | 65.58 | 54.04 | 44.05 | 36.71 | 24.64 |
| shot | 64.83 | 54.97 | 47.32 | 35.17 | 24.26 |
| impulse | 65.32 | 57.01 | 47.40 | 35.29 | 28.20 |
| defocus | 65.11 | 54.15 | 42.78 | 36.00 | 23.80 |
| glass | 64.06 | 54.62 | 42.62 | 30.01 | 23.17 |
| motion | 64.00 | 54.06 | 43.82 | 34.58 | 21.41 |
| zoom | 61.09 | 51.24 | 42.87 | 31.93 | 20.06 |
| snow | 63.40 | 57.36 | 47.79 | 35.80 | 24.41 |
| frost | 62.37 | 52.55 | 46.55 | 34.92 | 23.44 |
| fog | 62.12 | 57.15 | 44.42 | 36.25 | 26.80 |
| bright | 62.94 | 56.35 | 46.72 | 34.40 | 25.53 |
| contrast | 63.93 | 56.04 | 44.92 | 39.09 | 27.84 |
| elastic | 63.53 | 53.33 | 43.38 | 35.26 | 21.50 |
| pixelate | 64.91 | 52.21 | 42.70 | 31.52 | 20.89 |
| jpeg | 63.81 | 53.60 | 42.42 | 31.03 | 22.07 |
| **Avg** | **63.80** | **54.58** | **44.65** | **34.53** | **23.87** |

## F    USE OF LLMS

LLM is used to aid or polish writing.

