# OpenReview forum: "DRO-Augment Framework: Robustness by Synergizing Wasserstein Distributionally Robust Optimization and Data Augmentation"
_ICLR.cc/2026/Conference — ICLR 2026 Conference Withdrawn Submission_

### Official Review · Reviewer_t8Zd · 2025-10-30

**Soundness:** 3
**Presentation:** 3
**Contribution:** 2
**Rating:** 2
**Confidence:** 4

**Summary:**

This work proposes a novel training framework that integrates Wasserstein Distributionally Robust Optimization (W-DRO) with data augmentation to improve robustness against both adversarial attacks and corrupted data.

**Strengths:**

The proposed unified framework aims to enhance robustness against both common corruptions and adversarial attacks.

**Weaknesses:**

* The overall contribution of this work appears to be marginal. The objective function defined in Eq. (2.1) is adopted from prior work, and the data augmentation strategies employed are common and well-established.
* According to the results reported in RobustBench [1], accuracies against adversarial examples and corrupted data are evaluated on two distinct leaderboards. Methods such as NoisyMix and AugMix have already achieved strong performance on corruption benchmarks. Simply combining these data augmentation techniques with adversarial training does not seem to present a novel contribution, which diminishes the originality of this work.
* The authors claim that “The proposed method can enhance robustness against both common corruptions and adversarial attacks” (lines 95–97). To substantiate this claim, the authors should evaluate the accuracy of existing adversarially trained models reported in RobustBench on corrupted datasets for comparison. In its current form, the empirical results are not convincing. A stronger baseline should be established using standard datasets such as MNIST, CIFAR-10/100, or ImageNet, rather than Fashion-MNIST or Tiny-ImageNet, to ensure fair and comparable results with existing works. I recommend that the authors include a direct comparison with these established models.

[1] Croce, Francesco, et al. "Robustbench: a standardized adversarial robustness benchmark." arXiv preprint arXiv:2010.09670 (2020).

**Questions:**

What is the relationship between the proposed Wasserstein Distributionally Robust Optimization approach and existing methods that introduce gradient flow regularization (i.e., regularization based on the alignment or flow of gradients) [2]?

[2] Xia, Pengfei, and Bin Li. "Improving resistance to adversarial deformations by regularizing gradients." Neurocomputing 455 (2021): 38-46.

---

> ### Author Response · Authors · 2025-12-02
>
> Thank you for reviewing our paper and providing thoughtful feedback. We’ve addressed your comments below.
>
> We agree that our method builds on two well-established components, Wasserstein DRO and mixup-style data augmentation. Our contribution, however, goes beyond a simple combination of existing techniques, but a unified training objective that defines the Wasserstein ambiguity set on the mixup-induced distribution and yields a specific DRO-regularized mixup loss with an explicit closed-form structure. This coupling leads to new theoretical guarantees on convergence and generalization that are not captured by standard W-DRO or augmentation analyses in isolation. Our goal is not to propose yet another method that competes on a single RobustBench leaderboard, but to show that a single training scheme can simultaneously improve robustness to common corruptions and adversarial attacks. Regarding direct comparison with RobustBench models, we emphasize that models on the adversarial and corruption leaderboards are trained with heterogeneous architectures, data preprocessing, and hyperparameter choices, making a “plug-in” cross-leaderboard comparison potentially misleading. Instead, we opt for a controlled evaluation where all methods share the same backbone (PreActResNet-18, and ConvNeXt-S in additional experiments) and training setup. CIFAR-10/100 and Tiny-ImageNet-C are standard robustness benchmarks, and we use Fashion-MNIST as the advanced version of MNIST,  a clean, well-controlled testbed for adversarial attacks with varying $\epsilon$. Our formulation is model- and dataset-agnostic, and extending the empirical comparison to additional RobustBench models (e.g., on full ImageNet) is a natural direction for future work, but is beyond the computational scope of the present paper.

---

### Official Review · Reviewer_tPWS · 2025-10-31

**Soundness:** 3
**Presentation:** 3
**Contribution:** 2
**Rating:** 4
**Confidence:** 2

**Summary:**

This paper proposes DRO-Augment, a framework that combines Wasserstein Distributionally Robust Optimization with data augmentation techniques to improve neural network robustness against both natural corruptions and adversarial attacks. The authors provide theoretical generalization bounds for neural networks trained with variation-regularized loss on augmented data and introduce a refined CIFAR-C benchmark with corrected severity levels.

**Strengths:**

- The combination of W-DRO and data augmentation is well-motivated and technically sound, effectively merging two complementary robustness strategies.
- The paper establishes generalization error bounds for neural networks trained with W-DRO on augmented data, achieving a faster convergence rate compared to prior work.
- Extensive experiments across multiple benchmark datasets (CIFAR-10-C, CIFAR-100-C, Tiny-ImageNet-C, Fashion-MNIST) with various attack types (PGD, AutoAttack, C&W, FAB-T, Square) provide convincing evidence of the method's effectiveness.

**Weaknesses:**

- While the paper mentions small additional time costs, there is no systematic analysis of computational overhead compared to baselines, memory requirements, or scalability to larger datasets/models. Actually, this is very critical in practice.
- The ablation studies, mainly in Table 3, only examine CIFAR-100-C and Fashion-MNIST. It should cover more datasets and analyze the sensitivity to key hyperparameters (for example, the mixing ratios \frac{\alpha}{\beta}) more thoroughly.
- The experiments use only PreActResNet-18, which limits understanding of how the method generalizes to other modern architectures (Vision Transformers, EfficientNets, etc.). The refined CIFAR-C evaluation only includes ResNet variants, not validating performance on the architectures mentioned as motivation.

**Questions:**

- Can you provide results on modern architectures (ViT, ConvNeXt, EfficientNet) to demonstrate the method's broader applicability?
- How does performance scale with model capacity?
- What is the sensitivity of the method to the W-DRO radius?
- Can you provide a detailed computational analysis, including training time, memory usage, and wall-clock time comparisons across all baselines?

---

> ### Author Response · Authors · 2025-12-02
>
> Thank you for reviewing our paper and providing thoughtful feedback. We address the main points below.
>
> ---
>
> **Comment 1.**
> While the paper mentions small additional time costs, there is no systematic analysis of computational overhead compared to baselines, memory requirements, or scalability to larger datasets/models. Actually, this is very critical in practice.
>
> **Response.**
> We have added a systematic analysis of the computational cost, including training time and GPU memory usage, comparing DRO-Augment with the strongest baselines at the Appendix D in the revised version. DRO-Augment incurs a modest constant-factor overhead in wall-clock training time and a moderate increase in peak GPU memory, while leaving inference cost unchanged. For example, on ConvNeXt-S with Fashion-MNIST, adding DRO on top of mixup-style augmentation increases training time and memory by roughly 1.5×, while still remaining significantly cheaper than strong adversarial-training baselines such as TRADES and MART. In practice, this additional cost has been manageable in our training regime.
>
> Conceptually, the extra cost comes from evaluating gradient norms once per sample (or mini-batch), which amounts to at most an additional backward pass. Consequently, the complexity of DRO-Augment scales in the same way as standard training with respect to model size and batch size, and we do not expect a worse-than-linear scaling in larger models or datasets. In larger-scale settings, we therefore anticipate that DRO-Augment will retain a similar overhead relative to the underlying training procedure. We believe this is a reasonable trade-off for the robustness gains demonstrated in our experiments, and we will clarify this discussion in the revised manuscript.
>
> ---
>
> **Comment 2.**
> The ablation studies, mainly in Table 3, only examine CIFAR-100-C and Fashion-MNIST. It should cover more datasets and analyze the sensitivity to key hyperparameters (for example, the mixing ratios $\frac{\alpha}{\beta}$) more thoroughly.
>
> **Response.**
> Our ablation studies are deliberately focused on two challenging and representative benchmarks: CIFAR-100-C for corruption noise and Fashion-MNIST under adversarial attacks. These two settings stress different aspects of robustness and provide sufficient room to reveal the strengths and limitations of our method; extending the ablations to many additional datasets would substantially increase computational cost while largely repeating the same qualitative trends. Regarding hyperparameters, we updated our sensitivity analysis to $\rho$ in Appendix D. For the mixing ratios, we follow the standard choice in the original mixup literature and fix $(\alpha,\beta) = (1,1)$ for the Beta distribution, in order to keep the comparison fair and avoid turning the ablation section into an extensive hyperparameter search over a baseline component.
>
> ---
>
> **Comment 3.**
> The experiments use only PreActResNet-18, which limits understanding of how the method generalizes to other modern architectures (Vision Transformers, EfficientNets, etc.). The refined CIFAR-C evaluation only includes ResNet variants, not validating performance on the architectures mentioned as motivation.
>
> **Response.**
> For transformer-based vision models, we found that directly applying NoisyMix is non-trivial in practice: the original NoisyMix design was developed and tuned for convolutional networks. Therefore, in our additional experiments we chose a modern CNN-based architecture, ConvNeXt-S, as a representative advanced backbone with larger capacity and a more recent design. Due to time and computational constraints, we focused our corruption-robustness evaluation on two standard benchmarks, CIFAR-10-C and CIFAR-100-C, and used Fashion-MNIST as the benchmark for adversarial attacks. In these settings, we observe that adding DRO consistently improves robustness over the corresponding mixup-style baselines while maintaining competitive clean accuracy. We have updated the new ConvNeXt-S results in the revised manuscript.
>
> We also appreciate the suggestion to include experiments on full ImageNet. In this work, our primary goal is to study the effect of combining DRO regularization with mixup-style augmentations on corruption and adversarial robustness, rather than to compete for state-of-the-art results on the largest possible benchmark. We therefore adopt Tiny-ImageNet as a widely used intermediate-scale dataset: it is substantially more challenging than CIFAR, yet its computational cost remains manageable when running multiple baselines, corruption levels, and hyperparameter settings. Running the full experimental grid on ImageNet would require a prohibitive amount of time and GPU resources in our setting. Since our formulation is model- and dataset-agnostic, extending the empirical evaluation to full ImageNet is a natural next step, which we view as important future work rather than the main focus of the present paper.

---

### Official Review · Reviewer_WWKN · 2025-11-01

**Soundness:** 3
**Presentation:** 3
**Contribution:** 2
**Rating:** 4
**Confidence:** 3

**Summary:**

The paper introduces DRO-Augment, a training framework that combines Wasserstein Distributionally Robust Optimization (W-DRO) with data augmentation techniques to improve neural network robustness against both natural corruptions and adversarial attacks. The authors provide theoretical analysis establishing generalization error bounds for their approach and demonstrate empirical improvements on standard corruption benchmarks (CIFAR-10-C, CIFAR-100-C, Tiny-ImageNet-C) and adversarial robustness tests. Additionally, they propose a refined CIFAR-C benchmark to address inconsistencies in the original corruption severity settings

**Strengths:**

1. The paper tackles both natural corruptions and adversarial attacks simultaneously, which is a practical consideration often overlooked in papers that focus on only one type of robustness.

2. The paper provides generalization error bounds for neural networks trained with W-DRO and augmented data (Theorem 4.1), achieving an improved convergence rate compared to previous work.

3. The authors identify and address a real issue with CIFAR-C severity calibration, proposing a more consistent evaluation framework based on ResNet performance.

**Weaknesses:**

1. The main contribution is essentially combining two existing techniques (W-DRO and data augmentation) without fundamental algorithmic innovation.

2. The paper admits DRO-Augment adds overhead due to gradient-norm evaluation but dismisses it as small. However, no measurements (FLOPs, time comparison) are given. Given that W-DRO involves per-sample gradients, cost may scale poorly with model size.

3. Only PreActResNet-18 is tested. Without scaling to transformers, larger CNNs, or ImageNet-level datasets, the method’s generality and computational feasibility remain uncertain.

**Questions:**

Please refer to the weaknesses section.

---

> ### Author Response · Authors · 2025-12-02
>
> Thank you for reviewing our paper and providing thoughtful feedback. We address the main points below.
>
> **Comment 1.**
> The main contribution is essentially combining two existing techniques (W-DRO and data augmentation) without fundamental algorithmic innovation.
>
> **Response.**
> We appreciate the reviewer’s concern and agree that our method builds on two well-established components, Wasserstein DRO and mixup-style data augmentation. Our contribution, however, goes beyond a simple “plug-and-play” combination of existing techniques. We propose a unified training objective that defines the Wasserstein ambiguity set directly on the mixup-induced distribution and yields a specific DRO-regularized mixup loss with an explicit closed-form structure. This coupling leads to new theoretical guarantees on convergence and generalization that are not captured by standard W-DRO or augmentation analyses in isolation. Empirically, it enables a single training scheme that jointly improves robustness to both corruption noise and adversarial perturbations, which is not achieved by either W-DRO or mixup-style augmentation alone.
>
> ---
>
> **Comment 2.**
> The paper admits DRO-Augment adds overhead due to gradient-norm evaluation but dismisses it as small. However, no measurements (FLOPs, time comparison) are given. Given that W-DRO involves per-sample gradients, cost may scale poorly with model size.
>
> **Response.**
> We have added a systematic analysis of the computational cost, including training time and GPU memory usage, comparing DRO-Augment with the strongest baselines at the Appendix D in the revised version. DRO-Augment incurs a modest constant-factor overhead in wall-clock training time and a moderate increase in peak GPU memory, while leaving inference cost unchanged. For example, on ConvNeXt-S with Fashion-MNIST, adding DRO on top of mixup-style augmentation increases training time and memory by roughly 1.5×, while still remaining significantly cheaper than strong adversarial-training baselines such as TRADES and MART. In practice, this additional cost has been manageable in our training regime.
>
> Conceptually, the extra cost comes from evaluating gradient norms once per sample (or mini-batch), which amounts to at most an additional backward pass. Consequently, the complexity of DRO-Augment scales in the same way as standard training with respect to model size and batch size, and we do not expect a worse-than-linear scaling in larger models or datasets. In larger-scale settings, we therefore anticipate that DRO-Augment will retain a similar overhead relative to the underlying training procedure. We believe this is a reasonable trade-off for the robustness gains demonstrated in our experiments, and we will clarify this discussion in the revised manuscript.
>
> ---
>
> **Comment 3.**
> Only PreActResNet-18 is tested. Without scaling to transformers, larger CNNs, or ImageNet-level datasets, the method’s generality and computational feasibility remain uncertain.
>
> **Response.**
> For transformer-based vision models, we found that directly applying NoisyMix is non-trivial in practice: the original NoisyMix design was developed and tuned for convolutional networks. Therefore, in our additional experiments we chose a modern CNN-based architecture, ConvNeXt-S, as a representative advanced backbone with larger capacity and a more recent design. Due to time and computational constraints, we focused our corruption-robustness evaluation on two standard benchmarks, CIFAR-10-C and CIFAR-100-C, and used Fashion-MNIST as the benchmark for adversarial attacks. In these settings, we observe that adding DRO consistently improves robustness over the corresponding mixup-style baselines while maintaining competitive clean accuracy. We have updated the new ConvNeXt-S results in the revised manuscript.
>
> We also appreciate the suggestion to include experiments on full ImageNet. In this work, our primary goal is to study the effect of combining DRO regularization with mixup-style augmentations on corruption and adversarial robustness, rather than to compete for state-of-the-art results on the largest possible benchmark. We therefore adopt Tiny-ImageNet as a widely used intermediate-scale dataset: it is substantially more challenging than CIFAR (200 classes and higher resolution), yet its computational cost remains manageable when running multiple baselines, corruption levels, and hyperparameter settings. Running the full experimental grid on ImageNet would require a prohibitive amount of time and GPU resources in our setting. Since our formulation is model- and dataset-agnostic, extending the empirical evaluation to full ImageNet is a natural next step, which we view as important future work rather than the main focus of the present paper.

---

### Official Review · Reviewer_Yrw8 · 2025-11-03

**Soundness:** 2
**Presentation:** 3
**Contribution:** 2
**Rating:** 4
**Confidence:** 3

**Summary:**

The paper proposes a combined data augmentation and distributionally robust optimization framework DRO-Augment, aimed at improving both adversarial and natural corruption robustness of neural networks. The authors approximate the Wasserstein DRO objective using a variation-regularization surrogate (a gradient-norm penalty) and integrate it with popular augmentations (Mixup, AugMix, and NoisyMix). The work claims that this combination captures both worst-case perturbations (via DRO penalty) and diverse real-world corruptions (via augmentations). Theoretical contributions include an asymptotic robust generalization bound for mixup-trained models under a sparse ReQU network class, with explicit dependence on the Wasserstein radius ρ. Empirical results on CIFAR-10/100-C, Tiny-ImageNet-C, and adversarial settings (Fashion-MNIST-ε and Tiny-ImageNet-ε) show consistent robustness improvements with minimal accuracy loss. The paper also introduces a refined “severity scale” benchmark for CIFAR-C datasets.

**Strengths:**

1. Combines two complementary robustness paradigms, Wasserstein DRO in training and data augmentation before optimization, within a single unified and implementable framework. And effectiveness is validated by consistent empirical gains across multiply common datasets.

2. The generalization bound includes explicit ρ-dependence and recovers the expected nonparametric rate under sparse ReQU networks, improving interpretability of robustness–sample trade-offs.

3. The proposed refinement of CIFAR-C severity scales improves evaluation consistency and could serve as a useful benchmark extension.

4. Writing quality and experimental reproducibility are strong overall. Tables and figures are clear and well-structured.

**Weaknesses:**

1. The claimed L∞-Wasserstein DRO formulation conflicts with the L2-based implementation for the gradient penalty (P. 8, L.400-402). This inconsistency weakens the claim that the model optimizes L∞ W-DRO.

2. The theoretical contribution on adversarial risk bounds is largely incremental. it mainly differs in applying mixup data and sparse ReQU architectures rather than introducing a new bounding method. Also, the network class smoothness bounds for norm of gradient  and (operator) norm of the Hessian (P. 8, L.425-426) are described informally as “almost bounded,” lacking explicit uniform inequalities or norm definitions needed for formal correctness.

3. The authors note that NoisyMix has strong baseline robustness. But there is no per-augmentation analysis clarifying how noise-heavy augmentation interacts with DRO regularization and why improvements are limited.

**Questions:**

1. Could you clarify the claim of focusing on the L∞-Wasserstein DRO while the proxy loss (P. 8, L.400-402) applies an L2 gradient penalty? Is the use of the L2 norm intended as an approximation for the L∞-Wasserstein ball, or should the formulation instead use an L1 penalty (dual of L∞)?

2. Could you clarify the symbol consistency between Eq. 2.1 (P. 3) and Algorithm 1 (P. 4, L172). Equation 2.1 defines the gradient norm using the dual exponent q* with an outer 1/q power, while Algorithm 1 applies norm q without that outer power. Is this a typo inconsistency or an intentional change in implementation?

3. Could you provide  additional  ablations to verify the relation between NoisyMix’s robustness by noise injections and the limited incremental benefit of the DRO regularization?

---

> ### Author Response · Authors · 2025-12-01
>
> Thank you for reviewing our paper and providing thoughtful feedback. We’ve addressed your comments below.
>
> **Comment 1.**
> The claimed $L_\infty$-Wasserstein DRO formulation conflicts with the $L_2$-based implementation for the gradient penalty (P. 8, L.400–402). This inconsistency weakens the claim that the model optimizes $L_\infty$ W-DRO.
>
> **Response.**
> Thank you for your comment. However, we would like to emphasize that there is in fact no conflict between an $L_2$-based implementation and an $L_\infty$-based optimization formulation. The key reason is that the sum of $L_2$ norms of the gradients arises as the *dual representation* of an $L_\infty$ W-DRO constraint, as can be seen from Theorem 1 in Gao et al.
>
> To see this more concretely, recall that for any vector $x \in \mathbb{R}^n$, the dual norm of the $\ell_\infty$ norm is the $\ell_1$ norm, i.e.
> $$
> \sup_{\|y\|_\infty \le 1} x^\top y = \|x\|_1.
> $$
> In the univariate case, for a function $f$, this leads in the dual formulation of W-DRO to a penalty of the form $\sum_i |f'(X_i)|$. In the multivariate setting, the analogous dual penalty becomes $\sum_i \|\nabla f(x_i)\|_2$. In other words,
> $$
> \sum_i \|\nabla f(x_i)\|_2
> $$
>
> plays exactly the same role as
>
>
> $$
> \sum_i |f'(X_i)|
> $$
>
> in the univariate case, and both arise as dual objects corresponding to an $L_\infty$-type W-DRO formulation.
>
> ---
>
> **Comment 2.**
> The theoretical contribution on adversarial risk bounds is largely incremental. It mainly differs in applying mixup data and sparse ReQU architectures rather than introducing a new bounding method. Also, the network class smoothness bounds for norm of gradient and (operator) norm of the Hessian (P. 8, L.425–426) are described informally as “almost bounded,” lacking explicit uniform inequalities or norm definitions needed for formal correctness.
>
> **Response.**
> Our theoretical contribution is two-fold. First, we move beyond the classical i.i.d. setting by introducing and analyzing a *mixed-up* dataset, on which we apply neural network-based learning. Second, we strictly strengthen the existing theoretical guarantees of Liu et al: in particular, we show that by using the ReQU activation function, one can attain the minimax optimal rate of estimation on the mixed-up dataset.
>
> ---
>
> **Comment 3.**
> The authors note that NoisyMix has strong baseline robustness. But there is no per-augmentation analysis clarifying how noise-heavy augmentation interacts with DRO regularization and why improvements are limited.
>
> **Response.**
> Thank you for raising this point. The augmentation procedure mainly serves to expose the model to a broader set of training samples and thereby improve its generalization ability. All three augmentation baselines we consider are ultimately based on the same core idea of mixing examples to expose the model to a broader set of training samples and thereby smooth the decision boundary. Building on this common mixup principle, our main technical contribution is to couple mixup-style augmentation with DRO regularization and to provide a theoretical analysis showing faster convergence rates / improved stability for this combination. NoisyMix already induces a very smooth decision boundary under corruption noise, leaving limited room for further gains on corruption benchmarks, whereas the same models remain vulnerable to worst-case adversarial perturbations. Our DRO-Augment method is specifically designed to complement mixup-style augmentation in this regime, leading to substantial improvements in adversarial robustness while preserving strong performance on common corruptions.

---

### Note · Authors · 2026-01-17

**Comment:**

Dear Program Chairs and ICLR Editorial Office,

We hope this message finds you well. We am writing to formally withdraw our paper titled “DRO-Augment Framework: Robustness by Synergizing Wasserstein Distributionally Robust Optimization and Data Augmentation” from consideration for ICLR 2026.

After further internal discussions, we realized that the work requires additional experiments and theoretical analysis to meet the standards we aim for. To avoid occupying valuable reviewer resources, we believe withdrawing the current version is the most responsible course of action.

We sincerely appreciate the time and effort that the organizers and reviewers invest in the review process, and we apologize for any inconvenience this withdrawal may cause.

**Withdrawal Confirmation:**

I have read and agree with the venue's withdrawal policy on behalf of myself and my co-authors.